# Urban Spatial Configuration and Functional Runoff Connectivity: Influence of Drainage Grid Density and Landscape Metrics

**Vincent Smets** [1,*]**, Boud Verbeiren** [2]**, Martin Hermy** [1] **and Ben Somers** [1]

1   Division Forest, Nature and Landscape, Department of Earth and Environmental Sciences, KU Leuven, Celestijnenlaan 200E, 3001 Heverlee, Belgium; Martin.hermy@kuleuven.be (M.H.); Ben.somers@kuleuven.be (B.S.)
2   Department of Hydrology and Hydraulic Engineering, VUB, Pleinlaan 2, BE-1050 Brussels, Belgium; Boud.verbeiren@vub.be
*   Correspondence: Vincent.smets@kuleuven.be; Tel.: +3216374853

**Abstract:** Due to changing precipitation patterns induced by climate change, urban planners are confronted with new challenges to effectively mitigate rainfall runoff. An important knowledge gap that needs to be addressed before tackling these challenges is how and to which extent street/drainage grid density and spatial land use configuration influence the amount of runoff. Therefore, a virtual experiment was conducted to assess the influence of grid density and spatial land use configuration on the functional runoff connectivity ($F_c$), which is a measure of the easiness by which water flows through the landscape. Through the use of a design of experiments approach in combination with the SCS—Curve Number runoff model, a wide variety of neutral landscape models with a fixed percentage of pervious- and impervious cover were generated that maximized the variance of $F_c$. Correlations between landscape metrics and neutral landscape models were calculated. Our results indicated that, out of the 17 landscape metrics tested, the average impervious cluster area, the number of impervious clusters, the standard deviation of the cluster size, two proximity indexes and the effective impervious area were strongly correlated with $F_c$ throughout all grid scenarios. The relationship between $F_c$ on the one hand and the average impervious cluster area and the effective impervious area on the other hand, was modelled. The average impervious cluster area models showed a relationship with $F_c$ that closely approximated a logarithmic function ($R^2$: 0.49–0.73), while the effective impervious area models were found to have a linear relationship with $F_c$ ($R^2$: 0.63–0.99). A dense grid was shown to cause a strong increase in $F_c$, demonstrating the effectiveness of an urban grid in channeling and removing runoff. Our results further indicate that fine-grained landscapes with a lot of small impervious clusters are preferred over course-grained landscapes when the goal is to reduce $F_c$. In highly urbanized landscapes, where the percentage of impervious area is high, small changes in landscape pattern could significantly reduce $F_c$. By using a downward hydrological modeling approach this research aims to bring more clarity to the underlying variables influencing $F_c$, rather than trying to generate realistic prediction values.

**Keywords:** urban runoff; functional runoff connectivity; grid; neutral landscape models; curve number model; landscape metrics; impervious surface; urban green; optimization

# 1. Introduction

## 1.1. Context

Migration from rural to urban areas is rising around the world and is predicted to further increase. Currently around 54% of the world's population is living in urban areas and this number is predicted to increase to 66% by 2050 [1]. This migration towards urban areas brings a wide array of new challenges and opportunities. Sustainable, integral water management certainly is one of those challenges that is rapidly gaining traction, especially now that rainfall distribution patterns are changing due to climate change [2]. These changing rainfall patterns lead to a change in runoff processes. Runoff or overland flow is an important component of the hydrological balance that causes many problems such as flooding paired with loss of human lives and high economic costs [3,4]. The disrupted natural hydrological cycle in urban areas, impeding infiltration and stimulating runoff, exacerbates these problems [5,6]. The conventional solution to these problems is to guide the excess water away from the city [7]. Urban environments are characterized by grids of street networks that channel this runoff through the drainage system running underneath them. Once this water enters the drainage system, it is usually immediately diverted away from the city so it is lost for other usages within the cities' boundaries [8,9]. Moreover, the relatively clean rainwater often puts unneeded pressure on the drainage system and water treatment processing facilities, causing them to overflow when their maximal capacity is reached [10]. A solution to this problem could be to introduce more pervious spots in urban areas, disconnected from the drainage network, through which runoff can infiltrate and replenish the often depleted groundwater table underneath cities. Introducing more greenery in urban areas seems to be a simple, cost-effective, and aesthetically pleasing way of achieving this goal. Besides intercepting considerable amounts of rainfall [11], the soil under vegetation is less compacted than the soil under other urban land uses, thereby facilitating natural infiltration [12].

## 1.2. Problem Statement

Insight into which variables or processes govern the amount of overland flow that is guided to the drainage system is thus needed to establish a sustainable water management policy. It is well established that the amount of runoff is directly related to the amount of impervious area in a hydrological catchment [13]. However, the influence of the pattern of impervious area on hydrological variables such as runoff is much less studied [14]. The spatial arrangement of impervious area could be equally important as the impervious area per se because of the influence of connectivity and patchiness on runoff generation [15].

To date, most scientific literature regarding spatial land use pattern and its influence on runoff focuses on natural dryland areas. In a field study of hillslope runoff in drylands in Alicante, Spain, Mayor et al. [16] found that landscapes with a strongly clustered, coarser pattern increased runoff. The coarser structures enhanced the overall connectivity of the landscape classes, thereby enhancing runoff through the impervious class. The landscape metric they used to express landscape connectivity was the average flow length from the source (runoff producing) cells to the catchment's outlet. Other research found similar results in drylands: Coarser landscapes enhance water- and sediment fluxes while landscapes with a fine-grained spatial land use configuration reduced them [17]. In a follow-up study, they confirmed their results in a simulation experiment [18]. By creating landscapes with different spatial land use configurations and comparing water- and erosion fluxes, they found that discharge was greater for the landscapes where the land use classes were coarsely clustered.

The main limitation of these studies, when applying these findings to an urban environment, is the absence of roads in natural dryland areas. Roads, together with the drainage system running underneath them, act as channels for runoff that enhance the connectivity between impervious surfaces. Alberti et al. [14] discerned roads as a key stressor in urbanized landscapes. The more roads, the higher the runoff reaching the catchment's outlet. Shields and Tague [19] studied vegetation water use in an urban area with roads, found the effective impervious surface area (EIA) to be the main predictive

variable, with the vegetation receiving less water when EIA increases. EIA is the amount of impervious area directly connected to the drainage system. A higher disconnected impervious area had a positive effect on vegetation water use, meaning that a higher percentage of total impervious area does not necessarily mean that urban vegetation receives less water. They concluded that lowering the EIA fraction and making more impervious area hydrologically disconnected from the drainage network, will have the largest impact on conserving rainwater in the city.

*1.3. Research Questions*

More knowledge is needed on how and to which extent different urban environments, marked by different street grid densities, influence landscape pattern and runoff. More specifically, this study defines two main research questions:

1.  Which variables or landscape metrics influence runoff in an urban environment, characterized by different street grid densities?
2.  To what extent can a landscape, constrained by a certain street grid density and percentage of impervious surface, minimize runoff and maximize infiltration?

In general, the goal of this study is to provide a better insight in the processes governing overland flow in an urban environment. This knowledge can be used to optimize the urban spatial configuration to improve runoff channeling to the pervious areas in the landscape and partially restore the natural hydrological cycle. Because the drainage system plays an important role in channeling runoff, urban centers were chosen with different grid densities to provide insight in the effect of drainage network density on overland flow. Considering drainage networks are difficult to acquire due to region- or country specific limitations, the street grid was chosen as a replacement of the drainage network.

## 2. Material and Methods

This section is divided in three main parts according to the workflow adopted in this study (Figure 1). The process to generate Neutral Landscape Models (NLMs) with different grid densities and spatial patterns will be explained first. Next, the landscape metric selection and modelling procedure is described. Finally, the optimization technique used in this research to find landscapes with spatial patterns where runoff is minimized and infiltration is maximized is explained.

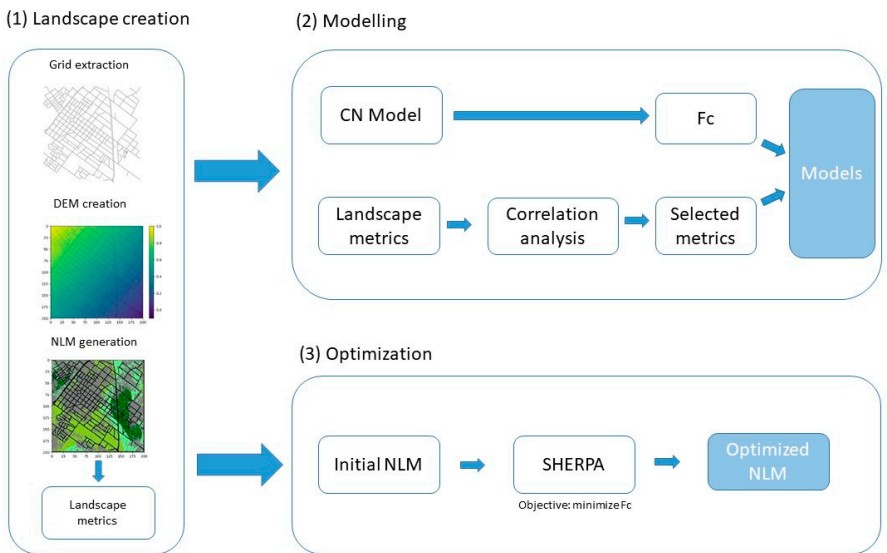

**Figure 1.** Schematic workflow of the study. (**1**) Landscape creation; (**2**) modelling and (**3**) Optimization. DEM: Digital Elevation Model; NLM: Neutral Landscape Models; CN model: Curve number model; $F_c$: Functional runoff connectivity; SHERPA: Optimization algorithm. The meaning of these terms will be explained in the remainder of the Material and Methods section.

*2.1. Landscape Creation*

2.1.1. Grid Extraction

Three urban grids were used in this study: (1) Irvine, California (4.2 km roads/km$^2$), (2) Concord, San Francisco (13.7 km roads/km$^2$), and (3) Point Breeze, Philadelphia (25.4 km roads/km$^2$) (Figure 2). Four km$^2$ of these urban grids were downloaded from www.openstreetmap.org with the Osmnx python plugin [20]. These grids were selected from an analysis of 27,000 urban street networks in the USA [21] and were chosen because of their large difference in road density. A fourth grid consisted only of an outlet and one road in the lower- and right boundary of the landscape to channel all runoff to the outlet. This grid represented a natural, pre-construction landscape and acts as a reference.

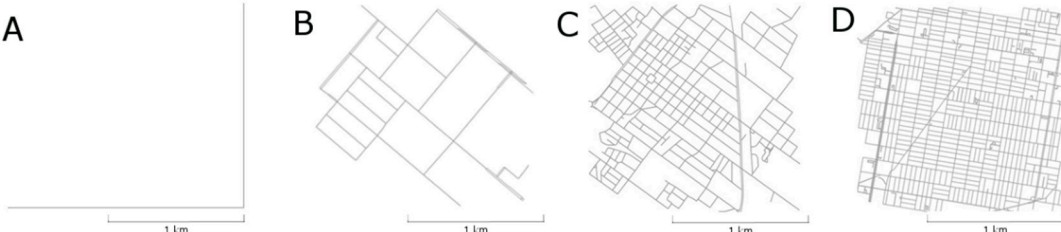

**Figure 2.** Urban grids (2 × 2 km) used in this study. (**A**) Reference scenario; (**B**) Irvine; (**C**) Concord; and (**D**) Point Breeze.

2.1.2. Digital Elevation Model Creation

A digital elevation model (DEM) was generated, sloping towards the lower-right corner of the landscape with the street grid embedded into it (Figure 3). This assured that, once runoff entered the grid, it remained there and was guided to the outlet. By generating an identical DEM for all NLMs, the virtual experiment was set up to optimally detect the influence of the grid and the spatial configuration of the landscape on runoff rather than to predict realistic numbers. Every landscape had only one outlet in the lower right corner where the outlet volume was used to calculate the runoff.



**Figure 3.** Example of a generated DEM with the grid of the Concord scenario embedded into it. The outlet is located in the lower right corner. *x*- and *y*-axis are in pixels (pixel size 10 × 10 m). The colors in the figure range from yellow (high) to dark blue (low). Because the grid is embedded in the landscape, runoff entering the grid will remain there until it reaches the outlet.

2.1.3. Neutral Landscape Model Generation

NLMs are able to generate spatial landscapes with different layers of complexity [22]. The number of land use classes, their abundance and complexity can be adapted according to the modeler's interests. Within the chosen urban grids, 200 × 200 pixel NLMs with a spatial resolution of 10 × 10 m were generated with the pseudo- random Perlin noise function. Perlin noise was originally conceived in

1983 to reproduce natural form and behavior. Since then it has been modified and used in a wide variety of applications, among which 2- and 3-dimensional realistic landscape generation [23]. It is also called fractal noise because of its addition of multiple noise layers with different weights that have varying amplitudes and wavelengths. Each layer is called an octave and the addition of these octaves results in a fractal structure that forms a natural, irregular pattern [24]. The more octaves are added to the function, the more fractal, or heterogeneous the final pattern will be. Different landscapes can by produced by Perlin noise by varying the number of octaves and several other intrinsic function variables. The fractal patterns produced by Perlin noise are suitable to reproduce urban landscapes because of the different layers of heterogeneity or organization inherent to them [25]. For example, urban areas consist of different land uses such as a commercial district or a park. Both land uses predominantly consist of impervious- and pervious respectively, but not exclusively. In the commercial district there will be small patches of pervious surface such as street trees or other small green elements, while in the park there will be small impervious patches such as a basketball field or a public bathroom.

　　　Urban land use complexity was downscaled by generating NLMs with a discrete number of land use classes: An impervious class consisting of impervious- and street grid surfaces, and a pervious class consisting of common urban green types. The pervious class contained grass, shrubs, and trees. On a first level of complexity, NLMs were constructed where the pervious and impervious surfaces varied in abundance. Landscapes where the pervious class occupied 20%, 40%, 60%, and 80% of the total surface area were generated, representing a transition from a very dense to a less dense urban area. In the high grid density Point Breeze scenario, the road grid consisted of 29.5% impervious surface, meaning a degree of 80% perviousness could not be reached. In the Concord scenario, the road grid occupied 18.9% of impervious surface, leaving only 1.1% impervious surface to be spatially allocated by the Perlin Noise function. For this reason, the highest level of pervious surface generated for these scenarios was 60%. A second level of complexity was added by applying the Perlin Noise function on the pervious surface area only. This way different spatial configurations of the above described urban green types were generated inside the pervious surface class. Proportions of the urban green types inside the pervious class were kept constant to accurately examine the effect of spatial urban green configuration on the runoff. The pervious surface consisted of 50% grass surface, 30% trees and 20% shrubs (Figure 4). The tree class contained both free standing- and more clustered park trees, as well as street trees standing directly besides a road.

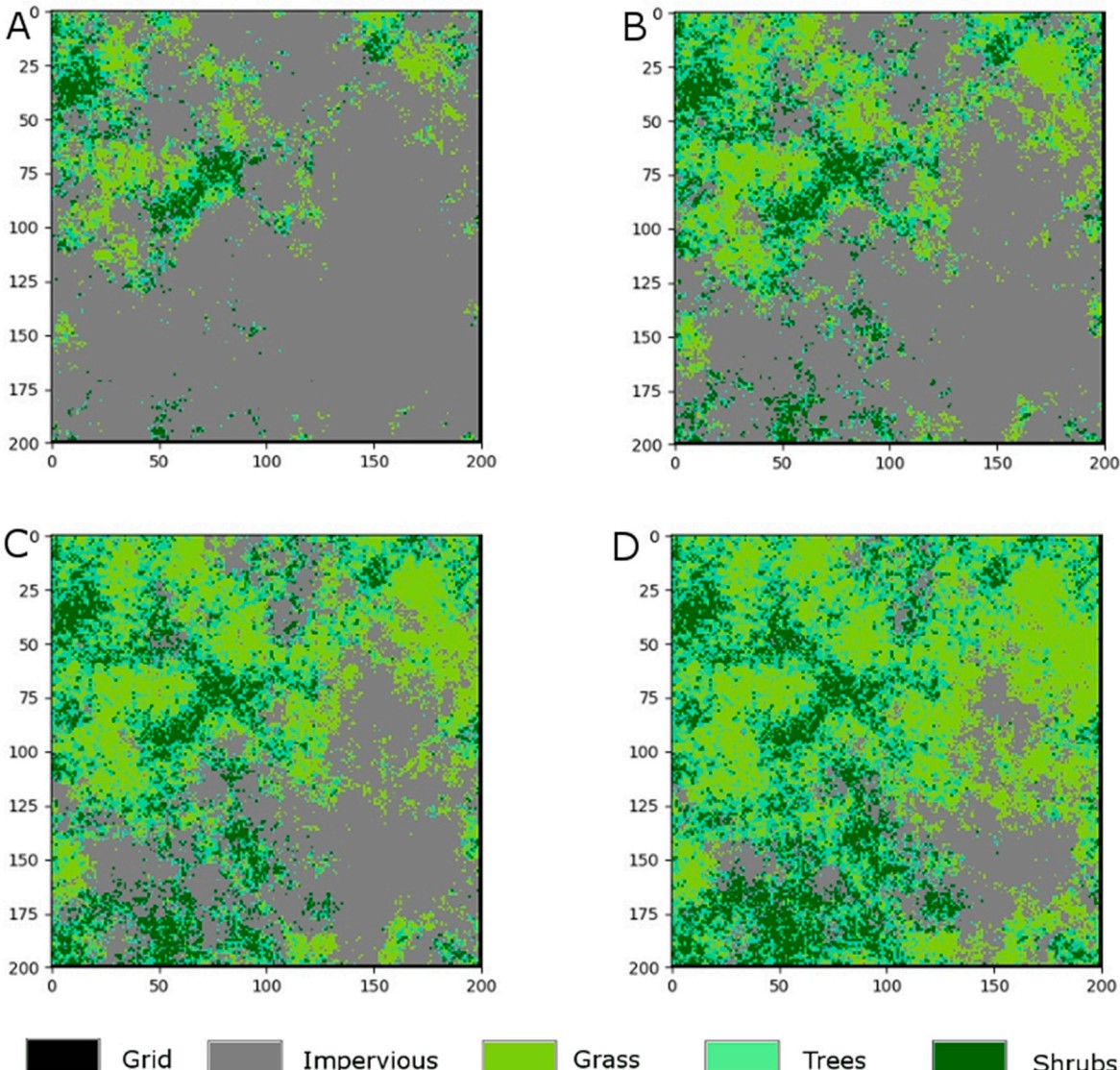

**Figure 4.** Example of an NLM with (**A**) 20%, (**B**) 40%, (**C**) 60%, and (**D**) 80% pervious surface in the Reference scenario.

### 2.1.4. Landscape Metrics

Landscape pattern is investigated through the usage of several landscape metrics. Landscape metrics were originally developed to measure the ecological impacts of landscape changes [26] but have since been applied to several other domains such as water quality assessment, urban sprawl detection and even to visual landscape appreciation [27]. To explain the observed differences in runoff between NLMs, the landscape metrics were calculated for each NLM (Table 1). A spatial clustering algorithm using a 4-neighbourhood classification divided the pervious- and impervious surface fraction in clusters that were used to calculate several landscape metrics. A cluster or patch is a basic element or unit that makes up a landscape [26]. In this study a cluster is defined as a unit of uniform land cover.

**Table 1.** Landscape metrics calculated for each NLM.

| Variable | Unit | Abbr. |
|---|---|---|
| Average area of impervious clusters | $m^2$ | A |
| Number of impervious clusters | no. | $N_{IC}$ |
| Number of pervious clusters | no. | $N_{PC}$ |
| Standard deviation of impervious cluster size | $m^2$ | $STD_{IC}$ |
| Size of the median impervious cluster | $m^2$ | $M_{IC}$ |
| Size of the largest impervious cluster | $m^2$ | $L_{IC}$ |
| Ratio of the effective impervious area to the total impervious area | / | EIA |
| Average flow path to outlet for each impervious pixel | m | $FP_{OIP}$ |
| Average flow path to nearest road for each impervious pixel | m | $FP_{RIP}$ |
| Average flow path from mass center impervious clusters to outlet | m | $FP_{MICO}$ |
| Average flow path from mass center impervious clusters to nearest road | m | $FP_{MICR}$ |
| Flow path largest impervious cluster to outlet | m | $FP_{LICO}$ |
| Flow path largest impervious cluster to nearest road | m | $FP_{LICR}$ |
| Proximity index impervious clusters to outlet | / | $PX_{ICO}$ |
| Proximity index impervious clusters to nearest road | / | $PX_{ICR}$ |
| Proximity index largest impervious cluster to outlet | / | $PX_{LICO}$ |
| Proximity index largest impervious cluster to nearest road | / | $PX_{LICR}$ |

The landscape metrics can be divided into area metrics, distance metrics, and proximity indexes, the latter of which are a combination of the first two [28]. The number of pervious clusters ($N_{PC}$) is not directly correlated with the number of impervious clusters ($N_{IC}$). The metric $N_{PC}$ calculates the number of pervious urban green patches and represents the spatial autocorrelation of urban green. A high $N_{PC}$ means that there are a lot of urban green patches and the urban green types are predominantly mixed. A low $N_{PC}$ means that the spatial configuration of the different green types is more coarse-grained and that all grass, shrub, or green patches are more clustered together. The EIA is calculated with the Flow Distance function in ArcPy. This function calculates the length of the flow path of all impervious pixels to the nearest road. Impervious pixels where the flow path does not reach a road are assigned a 'no data' value. The EIA is calculated as the ratio of impervious pixels of which the flow path reaches a road over the total number of impervious pixels. The landscape metrics that use the largest impervious cluster ($L_{IC}$, $FP_{LICO}$, $FP_{LICR}$, $PX_{LICO}$, $PX_{LICR}$) will be used to indicate if this cluster has a dominant effect on runoff in comparison with the other clusters. The proximity index (PX) gives an indication of the relative importance of a cluster in relation to other clusters. It is calculated as the ratio of the impervious clusters' sizes to the clusters' flow path to the outlet or road respectively. In this research the PX is used to describe the contribution of an impervious cluster to the amount of runoff that reaches the outlet. For example, when a small impervious cluster close to the outlet has the same contribution to the amount of runoff that reaches the outlet as larger impervious cluster that is further away from the outlet, their PX is the same. $PX_{ICO}$ and $PX_{ICR}$ are the ratios of the impervious clusters' sizes to the clusters' flow path to the outlet or road respectively. $PX_{LICO}$ and $PX_{LICR}$ are similar to $PX_{ICO}$ and $PX_{ICR}$ but only take the largest impervious clusters' size and flow path. In the Reference scenario, only landscape metrics were used that not contained the road grid. The metrics applicable to the Reference scenario are A, $N_{IC}$, $N_{PC}$, $STD_{IC}$, $M_{IC}$, $L_{IC}$, EIA, $FP_{OIP}$, $FP_{MICO}$, $FP_{LICO}$, $PX_{ICO}$, $PX_{LICO}$.

*2.2. Modelling*

2.2.1. Downward Modelling Approach

In the past decennia urban hydrological modeling has predominantly focused on ever increasingly refined, more detailed models to better predict urban water flows. This is the so-called reductionist approach. However, because of the complexity of the hydrological cycle and its multiple paths interacting at various scales, there is a limit to the extent the reductionist approach can accurately predict water flows, and more complex models inevitably have a higher uncertainty [29,30]. Moreover,

the success of these models in predicting the hydrological balance can to some extent be attributed to the extensive calibration and parametrization they require. This can lead to over-parameterization until the model accurately simulates reality. Because parameter calibration suffers from equifinality or the principle that a given result can be achieved through multiple pathways [31], the underlying real causes of the modelled variables' variation can be obscured. Opposite of the reductionist approach stands the downward approach [32]. This approach starts from a simple framework and its complexity can be increased in a hierarchical manner. Benefits of the downward approach include, according to [32], that individual model components can be selected, left out or combined based the modelers interest and that this approach is suited for all model types irrespective of their spatial resolution or degree of heterogeneity. This approach is thus not so much suitable for realistic prediction of the hydrological components, but serves well to increase our theoretical understanding of the processes at play that influence the observed variation. Due to the inherent complexity of an urban environment, the reductionist approach is bound to run into some problems. The downward approach, by scaling down the level of complexity present in an urban environment and by then subsequently adding layers of heterogeneity in a hierarchical matter, is better suited to provide insights in the causational processes of overland water flow [32].

Combining the downward hydrological modeling approach with NLMs provides an ideal experimental setup for testing the influence of grid density and spatial configuration of land uses on overland flow.

### 2.2.2. Curve Number Hydrological Model and Functional Runoff Connectivity

The SCS-Curve Number hydrological model (CN model) [33] was used to model overland flow. This event-based conceptual model assigns a number to each land-use class that represents its hydrological characteristics. Its dependence on a single parameter (CN number) makes the model easy to use and is often the preferred method for water conservation planning and flood control design [34]. The CN number essentially reduces the total precipitation to runoff, after abstraction of the other water balance components. To determine the CN number of a land use class, the hydrological soil group (HSG) and antecedent moisture condition (AMC) need to be specified. Because the HSG and AMC are undefined in this virtual experiment, their default values are used. The HSG of all land uses in this experiment is 'A' and their AMC is 'II'. For further elaboration on these variables, see [33]. The original CN model uses an initial abstraction equation before overland flow begins of $I_a = 0.2\,S$, where $I_a$ is the initial abstraction (mm) and $S$ is the potential water storage (mm) of a land use class. This study uses the equation $I_a = 0.05\,S$ which is based on more recent experiments that were fitted to urban environments [35]. The original curve numbers as developed by the SCS [36] are converted using the equations of [35]. The curve numbers used in the hydrological modelling procedure were 97.9 (grid), 85.5 (impervious), 33.7 (grass), 27.8 (trees), and 20.7 (shrubs). The CN model framework was originally developed for agricultural purposes and does not provide exact numbers for these urban green classes so some degree of liberty was taken in selecting those numbers. Because the goal of this research is to expose the influences of landscape pattern and grid on runoff instead of making realistic predictions, these values are deemed acceptable.

The functional runoff connectivity ($F_c$) (Equation (1)) was used to quantify the effect of grid density and spatial configuration on runoff. The $F_c$ is a dimensionless property of a system that is defined as the ratio of the total outlet volume over the internal runoff production and can be seen as the ease of which water flows through the landscape and reaches the outlet [28]. $F_c$ varies between 0 and 1, with 0 meaning all produced runoff infiltrates, while 1 means all produced runoff reaches the outlet. This variable was chosen because it is easily interpretable and comparable between scenarios because of its dimensionlessness.

$$F_c = \frac{Total\ outlet\ volume}{Total\ internal\ run\,off\ production} \tag{1}$$



Next, a rain event of 20 mm was simulated. Because the functional runoff connectivity is measured at the outlet and is not affected by rain duration or intensity, these rain event characteristics were not specified. Van Nieuwenhuyse et al. [28] found that the influence of pattern on runoff was most pronounced when the parameters of the different land covers differed sufficiently but not extremely, and that the simulated rainfall event made one type of land cover act as a source area and the other type as a sink area. A 20 mm rain event made the impervious areas act as a source patch, while the pervious areas acted as sink patches, absorbing runoff from nearby impervious pixels before getting saturated. During a rain event of 20 mm, the street grid and impervious surface produced 15.5 and 5.2 mm of runoff respectively, while no runoff was generated on the grass-, tree-, and shrub patches. During and after the rain event, these patches could absorb an additional 5, 13, and 28.7 mm of runoff respectively from nearby impervious surfaces before saturating.

### 2.2.3. Landscape Metric Correlation and Selection

To find the landscape metrics most heavily correlated with $F_c$, a process optimization software HEEDS|MDO 2019 [37] was used to perform a design of experiment (DOE) analysis. A design of experiments analysis is a method that examines how a selected number of variables influence a specified design. The Perlin Noise function variable values were selected with Latin hypercube sampling, generating a near-random sample of variable values to produce landscapes [38]. Next, for each grid and each percentage of pervious surface, 100 iterations were done. Spearman rank correlation plots were made of $F_c$ with the landscape metrics for each degree of perviousness. Spearman was preferred over Pearson because the Spearman's rank correlation coefficient describes monotonic relationships instead of linear relationships. Moreover Spearman makes no assumption about the underlying data distribution and Spearman's rho is more robust to outliers [39].

### 2.2.4. Landscape Metric and Functional Runoff Connectivity Modelling

The landscape metrics that were consistently correlated for all grid scenarios and all degrees of pervious surface were closer examined. A PCA and Spearman correlation analysis was done between these selected metrics to determine their level of similarity. Finally, the two most suitable landscape metrics were selected and their behavior in relation to $F_c$ was modelled. Model assumptions were checked and the goodness of fit ($R^2$) and root mean square error ($R_{MSE}$) were calculated.

### 2.3. Optimization with SHERPA Algorithm

To minimize $F_c$ for a given grid and percentage of impervious surface, HEEDS|MDO 2019 uses an optimization technique called SHERPA [40]. SHERPA is a hybrid adaptive method that satisfies the specified objective(s) by iteratively searching for more optimal landscape patterns. Starting from a user defined initial landscape pattern, SHERPA is able to find the NLM that best satisfies the specified objective (minimize $F_c$) faster than other methods by looking at previously produced landscape patterns. By specifying the surface area of all landscape classes as a constraint for the optimizing algorithm, SHERPA found the most optimal NLM for each grid with 20%, 40%, 60%, and 80% pervious surface area. The process of finding the best landscape pattern is depicted in Figure 5. Starting from an initial landscape pattern, SHERPA iteratively searches until it finds the landscape pattern best complying to the specified objective(s) and constraints.

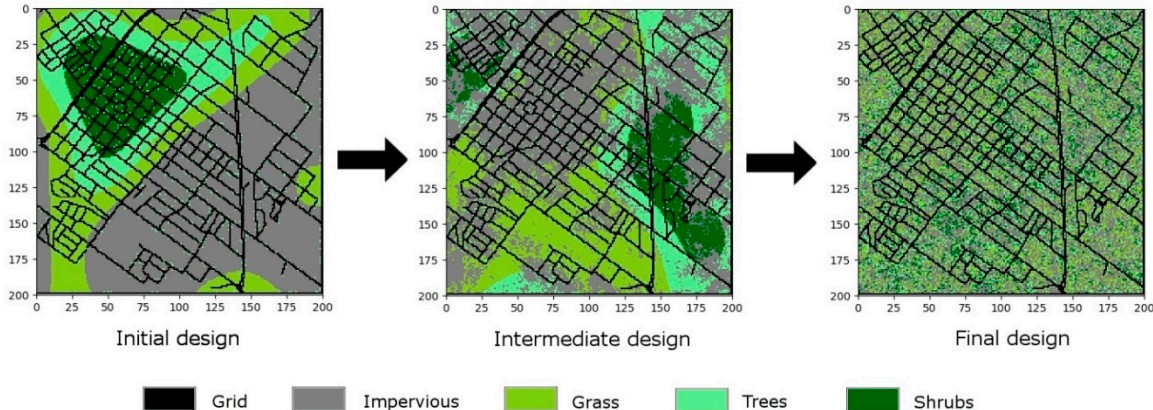

**Figure 5.** Example of optimizing method SHERPA: In this example the Concord grid is used with a pervious area percentage of 40%. The $F_c$ in the initial, intermediate and final landscape pattern is 0.97, 0.87, and 0.77 respectively.

## 3. Results

### 3.1. Landscape Metrics Correlations and Models

#### 3.1.1. Landscape Metrics Correlations

The Spearman correlations of the landscape metrics with $F_c$ are shown in Table 2. Notice that the Reference scenario has less landscape metrics because, due to the absence of a road grid, the metrics $FP_{RIP}$, $FP_{MICR}$, $FP_{LICR}$, $PX_{ICR}$, and $PX_{LICR}$ were not applicable.

**Table 2.** Spearman correlations of the landscape metrics for each degree of perviousness in the landscape. A deep blue color means a strong positive correlation while a deep red color signifies a strong negative correlation. See Table 1 for the full names of the landscape metrics. Extended tables with $p$ values and sample sizes can be consulted in Appendix A.

| | Pervious Fraction | | | |
|---|---|---|---|---|
| Reference | 0.20 | 0.40 | 0.60 | 0.80 |
| A | 0.40 | 0.76 | 0.78 | 0.35 |
| $N_{IC}$ | −0.40 | −0.76 | −0.78 | −0.35 |
| $N_{PC}$ | −0.74 | −0.63 | −0.45 | −0.12 |
| $STD_{IC}$ | 0.40 | 0.80 | 0.81 | 0.34 |
| $M_{IC}$ | 0.66 | 0.50 | 0.36 | 0.30 |
| $L_{IC}$ | −0.09 | 0.69 | 0.77 | 0.30 |
| EIA | 0.66 | 0.82 | 0.88 | 0.80 |
| $FP_{OIP}$ | −0.43 | −0.36 | −0.39 | −0.64 |
| $FP_{MICO}$ | 0.39 | 0.24 | 0.39 | −0.18 |
| $FP_{LICO}$ | −0.18 | −0.19 | −0.24 | −0.38 |
| $PX_{ICO}$ | 0.38 | 0.73 | 0.78 | 0.35 |
| $PX_{LICO}$ | 0.59 | 0.72 | 0.88 | 0.53 |
| | Pervious fraction | | | |
| Irvine | 0.20 | 0.40 | 0.60 | 0.80 |
| A | 0.69 | 0.90 | 0.93 | 0.93 |
| $N_{IC}$ | −0.69 | −0.90 | −0.93 | −0.93 |
| $N_{PC}$ | −0.78 | −0.60 | −0.38 | −0.15 |
| $STD_{IC}$ | 0.68 | 0.93 | 0.96 | 0.96 |
| $M_{IC}$ | −0.01 | −0.08 | 0.05 | 0.21 |
| $L_{IC}$ | −0.03 | 0.88 | 0.92 | 0.94 |
| EIA | 0.79 | 0.97 | 0.97 | 0.97 |
| $FP_{OIP}$ | −0.05 | −0.08 | 0.01 | 0.15 |
| $FP_{RIP}$ | −0.08 | 0.88 | −0.24 | −0.33 |

**Table 2.** *Cont.*

| Irvine | Pervious fraction | | | |
|---|---|---|---|---|
| | 0.20 | 0.40 | 0.60 | 0.80 |
| $FP_{MICO}$ | 0.10 | −0.08 | −0.06 | 0.06 |
| $FP_{MICR}$ | −0.45 | −0.29 | −0.83 | −0.75 |
| $FP_{LICO}$ | 0.01 | −0.20 | 0.02 | 0.17 |
| $FP_{LICR}$ | 0.22 | 0.04 | −0.19 | 0.14 |
| $PX_{ICO}$ | 0.64 | 0.88 | 0.93 | 0.94 |
| $PX_{ICR}$ | 0.70 | 0.89 | 0.93 | 0.93 |
| $PX_{LICO}$ | 0.21 | 0.67 | 0.89 | 0.89 |
| $PX_{LICR}$ | 0.07 | 0.71 | 0.84 | 0.85 |
| **Concord** | **Pervious Fraction** | | | |
| | 0.20 | 0.40 | 0.60 | 0.80 |
| A | 0.85 | 0.95 | 0.94 | - |
| $N_{IC}$ | −0.85 | −0.95 | −0.94 | - |
| $N_{PC}$ | −0.77 | −0.56 | −0.27 | - |
| $STD_{IC}$ | 0.85 | 0.96 | 0.93 | - |
| $M_{IC}$ | 0.83 | 0.80 | 0.83 | - |
| $L_{IC}$ | 0.39 | 0.88 | 0.86 | - |
| EIA | 0.80 | 0.98 | 0.98 | - |
| $FP_{OIP}$ | −0.11 | −0.07 | 0.06 | - |
| $FP_{RIP}$ | 0.04 | −0.09 | −0.14 | - |
| $FP_{MICO}$ | −0.03 | 0.16 | 0.30 | - |
| $FP_{MICR}$ | 0.40 | −0.69 | −0.88 | - |
| $FP_{LICO}$ | −0.18 | 0.07 | 0.08 | - |
| $FP_{LICR}$ | 0.05 | 0.19 | 0.29 | - |
| $PX_{ICO}$ | 0.75 | 0.94 | 0.94 | - |
| $PX_{ICR}$ | 0.85 | 0.94 | 0.93 | - |
| $PX_{LICO}$ | 0.27 | 0.67 | 0.79 | - |
| $PX_{LICR}$ | 0.38 | 0.48 | 0.63 | - |
| **Point B.** | **Pervious Fraction** | | | |
| | 0.20 | 0.40 | 0.60 | 0.80 |
| A | 0.93 | 0.97 | 0.97 | - |
| $N_{IC}$ | −0.93 | −0.97 | −0.97 | - |
| $N_{PC}$ | −0.77 | −0.50 | −0.13 | - |
| $STD_{IC}$ | 0.88 | 0.93 | 0.92 | - |
| $M_{IC}$ | 0.83 | 0.91 | 0.88 | - |
| $L_{IC}$ | 0.66 | 0.75 | −0.03 | - |
| EIA | 0.84 | 0.99 | 0.99 | - |
| $FP_{OIP}$ | 0.04 | 0.01 | 0.07 | - |
| $FP_{RIP}$ | 0.10 | −0.16 | −0.48 | - |
| $FP_{MICO}$ | −0.09 | −0.10 | 0.04 | - |
| $FP_{MICR}$ | 0.79 | 0.54 | −0.21 | - |
| $FP_{LICO}$ | −0.15 | 0.08 | 0.06 | - |
| $FP_{LICR}$ | −0.25 | −0.54 | 0.01 | - |
| $PX_{ICO}$ | 0.89 | 0.96 | 0.97 | - |
| $PX_{ICR}$ | 0.90 | 0.97 | 0.96 | - |
| $PX_{LICO}$ | 0.17 | 0.22 | 0.71 | - |
| $PX_{LICR}$ | 0.67 | 0.59 | 0.72 | - |

The most important observations from these correlation matrices are the following: A and $N_{IC}$ display behavior exactly opposite of each other throughout all scenarios and all percentages of perviousness. They are exact adversaries of each other because a higher average surface area of impervious clusters (A) means a lower number of impervious clusters ($N_{IC}$). A is positively correlated and $N_{IC}$ is negatively correlated with $F_c$, meaning that a high number of small impervious clusters results in a lower $F_c$ and a small number of large impervious clusters results in a higher $F_c$. This is also demonstrated in the high positive correlation of the standard deviation of the impervious clusters ($STD_{IC}$) throughout all scenarios and all percentages of perviousness. The strong positive correlation

of $STD_{IC}$ means that equal-sized impervious clusters are preferred over a wider variation of cluster sizes when the goal is to reduce $F_c$. Another metric that shows a strong positive correlation in all scenarios and degrees of perviousness is the effective impervious surface area (EIA). This means that a higher EIA invariably results in a higher $F_c$ and when the goal is to minimize $F_c$, care should be taken to reduce the impervious area directly connected to the street grid or outlet. For the Reference scenario, EIA depicts the amount of impervious surface directly connected to the lower- or right border of the landscape or outlet. Finally, $PX_{ICO}$ and $PX_{ICR}$ are two proximity indexes that show strong positive correlations throughout all scenarios and degrees of perviousness. Their positive correlation means that large impervious clusters close to the outlet or to a road result in a strong increase of $F_c$ and that if the goal is to reduce $F_c$, large impervious patches are best located far from the nearest road or outlet. Another measure that deserves attention is the number of pervious clusters ($N_{PC}$). In every grid scenario $N_{PC}$ starts with a strong negative correlation at a low percentage of perviousness, after which the strength of the correlation decreases with an increasing amount of pervious surface. This means that when the pervious surface area is small (e.g., 20%), a more mixed spatial configuration of urban green types is preferred to reduce $F_c$. When on the contrary a lot of pervious surface area is present (e.g., 80%) the spatial correlation of urban green does not matter much anymore.

The landscape metrics A, $N_{IC}$, $STD_{IC}$, EIA, $PX_{ICO}$, and $PX_{ICR}$ show consistent and strong behavior across all grid scenarios and all degrees of perviousness. After a PCA and Spearman analysis among these metrics, A, $N_{IC}$, $STD_{IC}$, $PX_{ICO}$, and $PX_{ICR}$ were shown to display strong levels of correlation, with only EIA showing distinct behavior. Out of these correlated metrics, the landscape metric A is selected to be modelled together with EIA due to its easiness to measure and intuitive understanding. The Spearman correlations of A and EIA with $F_c$ across all street grids and degrees of pervious surface are shown in Figure 6.

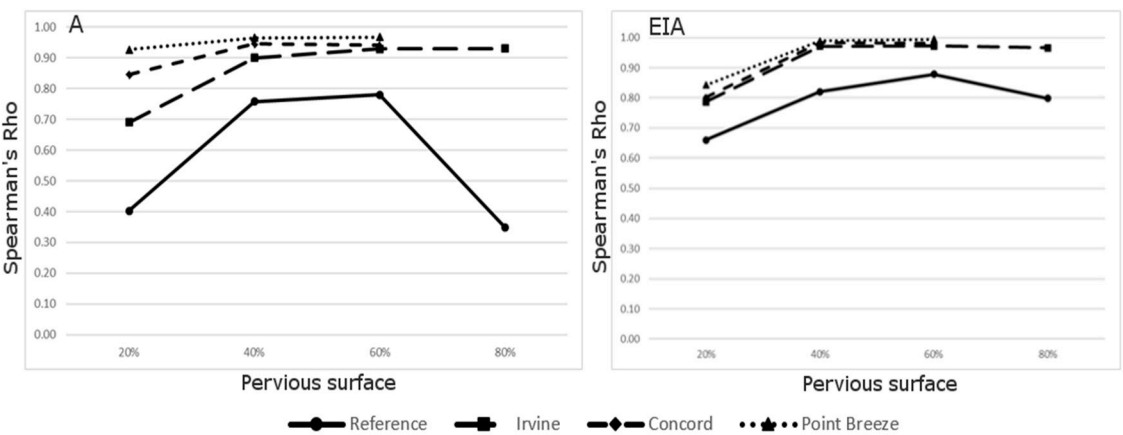

**Figure 6.** Spearman correlation plots for average impervious cluster area (A), and effective impervious surface area (EIA) with $F_c$ for each grid scenario across all degrees of pervious surface.

Some similar behavioral trends can be observed between the two landscape metrics plotted in Figure 6: the landscape metrics show consistently lower correlations in the Reference scenario than in the three other grid scenarios. Most notably the metric A still shows good correlations for the Reference scenario with a pervious area of 40% and 60% but in the more extreme scenarios (20% and 80% pervious surface area), the correlation coefficient drops significantly. The two metrics show good to very good correlations in the other grid scenarios with a slightly lower correlation when the pervious surface area is low (20%).

### 3.1.2. Landscape Metrics Models

When plotted with $F_c$, the landscape metric A followed a curve closely resembling a logarithmic function. The metric EIA could be modelled with a linear function. The variable A is modelled for

every grid scenario separately while EIA is modelled for every grid scenario and every degree of perviousness separately (Table 3).

**Table 3.** Models of $F_c$ with the variables A and EIA. The graphs of A can be viewed in Figure 7. The graphs of EIA can be viewed in Figures 8 and 9. Parameter estimates with 95% confidence intervals can be consulted in the Appendix B. The parameter a is the intercept of the functions, the parameter b is the base of the logarithmic function and the slope of the linear function.

| Metric | Model Function | Grid Scenario | Pervious (%) | Parameters | $R^2$ | $R_{MSE}$ | ANOVA |
|---|---|---|---|---|---|---|---|
| A | Logarithmic $F_c = a_i + b_i \times Log(A)$ | Reference | all | $a_1, b_1$ | 0.49 | 0.16 | <0.0001 |
| | | Irvine | all | $a_2, b_2$ | 0.60 | 0.10 | <0.0001 |
| | | Concord | all | $a_3, b_3$ | 0.73 | 0.03 | <0.0001 |
| | | Point Breeze | all | $a_4, b_4$ | 0.69 | 0.01 | <0.0001 |
| EIA | Linear $F_c = a_{ij} + b_{ij} \times EIA$ | Reference | all | $a_{11}, b_{11}$ | 0.85 | 0.09 | <0.0001 |
| | | Irvine | 20 | $a_{21}, b_{21}$ | 0.68 | 0.07 | <0.0001 |
| | | | 40 | $a_{22}, b_{22}$ | 0.96 | 0.04 | <0.0001 |
| | | | 60 | $a_{23}, b_{23}$ | 0.96 | 0.03 | <0.0001 |
| | | | 80 | $a_{24}, b_{24}$ | 0.90 | 0.03 | <0.0001 |
| | | Concord | 20 | $a_{31}, b_{31}$ | 0.63 | 0.03 | <0.0001 |
| | | | 40 | $a_{32}, b_{32}$ | 0.97 | 0.01 | <0.0001 |
| | | | 60 | $a_{33}, b_{33}$ | 0.97 | 0.01 | <0.0001 |
| | | Point Breeze | 20 | $a_{41}, b_{41}$ | 0.71 | 0.01 | <0.0001 |
| | | | 40 | $a_{42}, b_{42}$ | 0.99 | 0.00 | <0.0001 |
| | | | 60 | $a_{43}, b_{43}$ | 0.98 | 0.00 | <0.0001 |
| | | All grids | all | $a_5, b_5$ | 0.85 | 0.11 | <0.0001 |

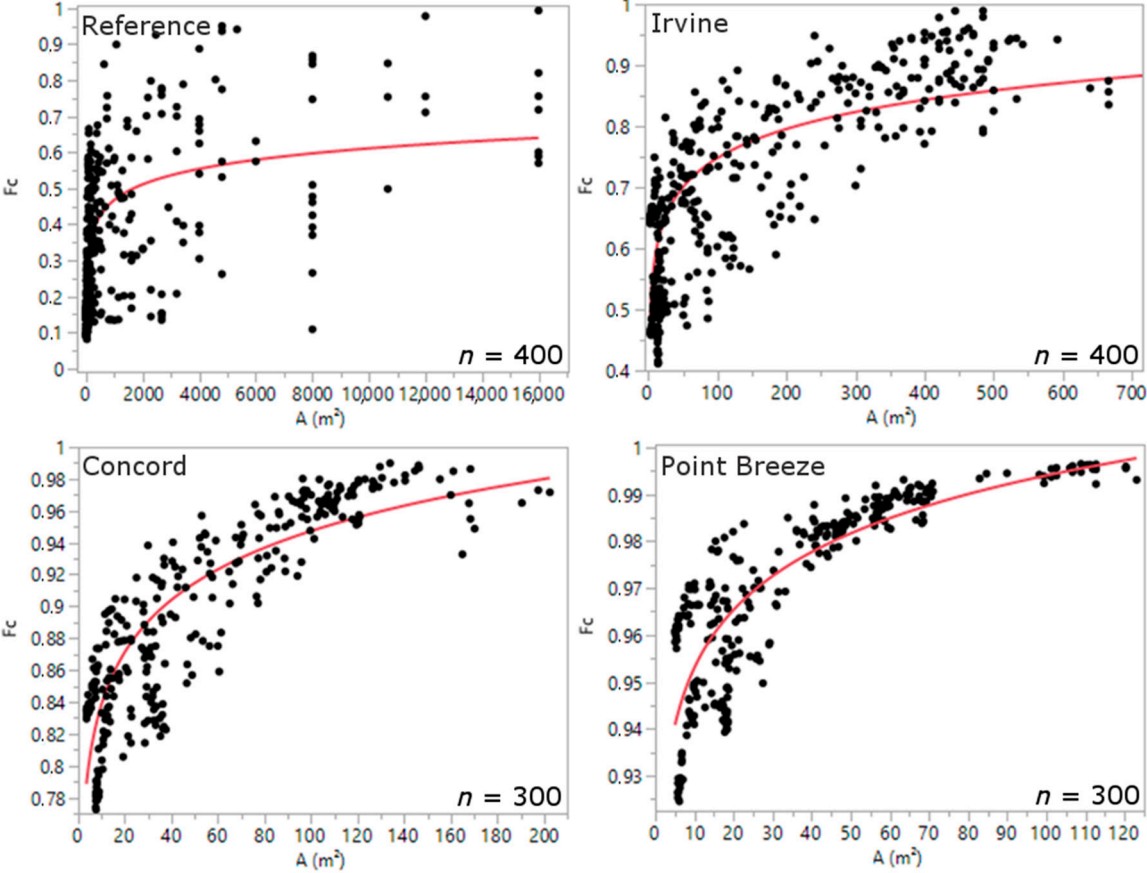

**Figure 7.** Scatterplots with logarithmic regression line for A and $F_c$ for the Reference, Irvine, Concord, and Point Breeze scenarios. Notice the different *x*- and *y*-axis in the figures. '*n*' is the number of NLMs generated for each grid scenario. Because in the Concord and Point Breeze scenarios no NLMs with 80% pervious surface were generated, these scenarios only have 300 NLMs.

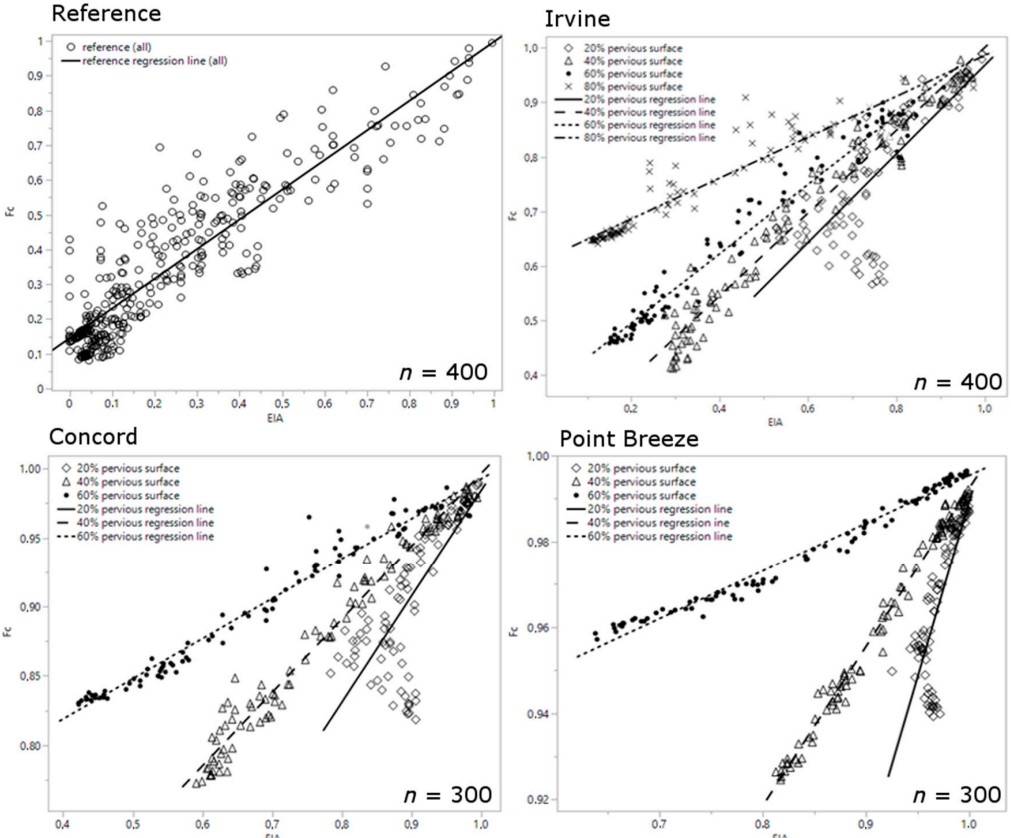

**Figure 8.** Scatterplots with linear regression lines for EIA and $F_c$ for the Reference, Irvine, Concord, and Point Breeze scenarios. The Reference scenario shows no difference between different levels of perviousness. The other scenarios do show marked differences between pervious percentages which are highlighted with different regression lines. '*n*' is the number of NLMs generated for each grid scenario. Because in the Concord and Point Breeze scenarios no NLMs with 80% pervious surface were generated, these scenarios only have 300 NLMs. Notice the different *x*- and *y*-axis in the figures.

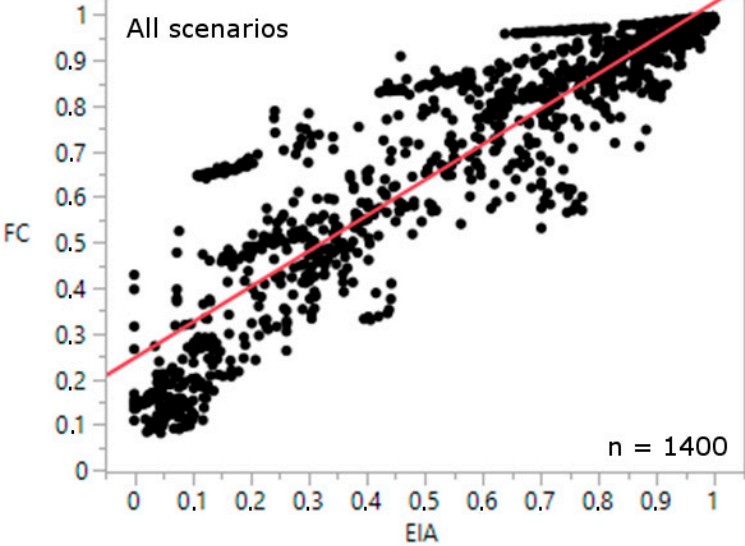

**Figure 9.** Scatterplot with linear regression line for EIA and $F_c$ for all four grid scenarios and all percentages of perviousness. '*n*' is the total number of NLMs generated for all grid scenarios together.

The parameters a and b are the intercept and base term of the logarithmic function. For the linear function the parameters a and b are the intercept and slope. In the case of A, different parameters were found for every grid scenario. The parameters with subscript '1', '2', '3', and '4' are the parameters for the Reference, Irvine, Concord, and Point Breeze scenario respectively. For the metric EIA, one linear model was constructed for the Reference scenario. For the Irvine, Concord, and Point Breeze scenarios, separate models were constructed for each degree of perviousness. The relationship between A and $F_C$ is shown in Figure 7 and the relationship between EIA and $F_C$ is shown in Figures 8 and 9. Because of the large differences in value ranges, the scales of the *x*- and *y*-axis of the subpanels in both Figures 7 and 8 are different. The parameter estimates with their 95% confidence intervals and standard errors can be found in Appendix B.

The metric A (Table 3 and Figure 7) shows a different behavior with increasing grid density. The higher the grid density, the more A fits a logarithmic curve when plotted against $F_c$. The $R^2$ increases from 0.49 in the Reference scenario to 0.73 and 0.69 in the Concord and Point Breeze scenarios while the $R_{MSE}$ decreases from 0.16 in the Reference scenario to 0.01 in the Point Breeze scenario. The street grid divides the landscape and this is reflected in the value of A. The range of the average impervious cluster area strongly decreases from the Reference to the Point Breeze scenario. The variation of the residuals seems to decrease slightly from low A values to high A values in all grid scenarios, indicating some degree of heteroscedasticity. This minor degree of heteroscedasticity however is not seen as a problem and corrections should only be made with more severe cases [41].

The variable EIA (Table 3, Figures 8 and 9) showed different behavior for different grid scenarios and degrees of perviousness. Only in the Reference scenario, one linear model could encompass all degrees of perviousness. With an $R^2$ of 0.85 and an $R_{MSE}$ of 0.09 this model succeeds very well in predicting $F_c$ for all degrees of perviousness when no roads are present in the landscape. When a grid is introduced in the landscape, model parameters change according to the degree of pervious surface. For each of the grid scenarios, the slope (parameter b) decreases when the degree of pervious surface increases. A minor increase in EIA will lead to a much larger increase in $F_c$ in a landscape with 20% pervious surface than in a landscape with 60% or 80% pervious surface. This effect is amplified with a higher grid density: In the Point Breeze scenario, the slope of 20% pervious regression line is steeper than the slope of the 20% pervious regression line in the Irvine scenario. A denser street grid also has influence on the possible range of values EIA can take. This range decreases strongly with higher grid density: In the Reference scenario EIA ranges from zero to one, while in the Point Breeze scenario EIA only ranges from 0.64 to one. As a result of the higher grid density in the Concord- or Point Breeze scenario, there are less isolated impervious clusters that are disconnected from the street grid, meaning EIA will have larger values.

A surprising observation is that in the three grid scenarios, at 20% pervious surface, the $R^2$ of the EIA models is considerably lower compared to other degrees of perviousness (Table 3: Irvine $R^2$: 0.67; Concord $R^2$: 0.63; Point Breeze $R^2$: 0.71). The relationship between EIA and $F_c$ in these cases shows a deviation from linearity and the lowest $F_c$ seems to be found at intermediate EIA values (Figure 8, 20% pervious regression line). At 20% pervious surface, $N_{PC}$ has a strong correlation with $F_c$ (Table 2), that quickly decreases with increasing degree of perviousness. When $N_{PC}$ is added as a predictor variable to these EIA models, the non-linearity instantly disappears and the resulting MLR reaches an $R^2$ comparable to the other EIA models (See Appendix C).

When all scenarios and all degrees of pervious surface are modelled together, a relatively good linear relationship could be modelled between $F_c$ and EIA with an $R^2$ of 0.85 and an $R_{MSE}$ of 0.11 (Figure 9).

*3.2. Optimization*

The lowest $F_c$ SHERPA found varied significantly between grids and pervious area abundances (Figure 10).

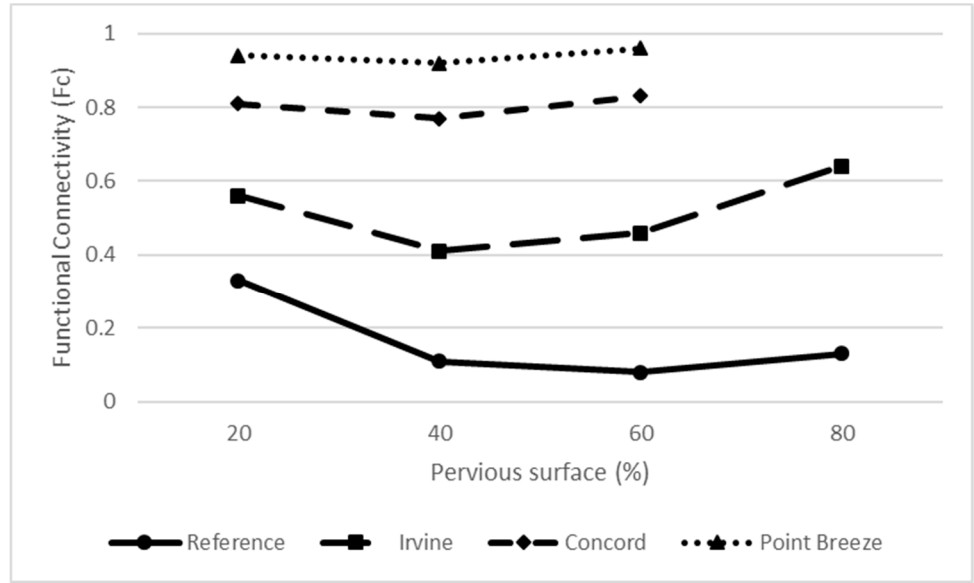

**Figure 10.** Lowest $F_c$ found by SHERPA for all grids and pervious surface abundances.

The first thing that stands out are the high differences in $F_c$ between the different grid scenarios. The higher the grid density, the higher $F_c$, demonstrating the efficiency of a grid transport system in channeling and leading runoff to the outlet. In all scenarios from 20% to 40% pervious surface $F_c$ decreases. This decrease in $F_c$ is greater in the low density grid scenarios (Reference and Irvine) than in the high density grid scenarios (Concord and Point Breeze). From 40% pervious surface upwards to 80% pervious surface, $F_c$, perhaps contra intuitively, rises again for the Irvine, Concord, and Point Breeze scenario.

## 4. Discussion

### 4.1. Landscape Metrics Correlations and Modelling

The focus of this virtual experiment was to determine landscape metrics that are strongly correlated with $F_c$ throughout a wide variety of landscapes. The average impervious cluster area (A), the number of impervious clusters ($N_{IC}$), the standard deviation of impervious clusters ($STD_{IC}$), the effective impervious area (EIA), the proximity index from the impervious clusters to the outlet ($PX_{ICO}$), and the proximity index from the impervious clusters to the nearest road (PXICR) were found to do this. Of these metrics A, $N_{IC}$, $STD_{IC}$, $PX_{ICO}$, and $PX_{ICR}$ showed strong correlations among each other, meaning they influence $F_c$ in a similar way. All these metrics have an impact on pattern formation in a landscape. The signs of their correlations with $F_c$ indicates that evenly distributed, fine-grained patterns are best suitable to reduce $F_c$ (Table 2). These findings are in line with [16–18]. Mayor et al. [16] used flow path length as a predictive variable to express landscape connectivity. This study could not find consistent correlations between flow path length and $F_c$ across all NLMs (Table 2) but the metrics $PX_{ICO}$ and $PX_{ICR}$, that include flow path length, did show a consistent correlation.

Figure 6 shows that with increasing street/drainage grid density, the correlation of these metrics with $F_c$ grows stronger. This indicates the importance of a grid in cluster formation and the dependence of these landscape metrics on grid density. As a denser grid produces more clusters, the strength of the correlations between the metrics and $F_c$ grows. This is also seen in Figure 7, where the relation between A and $F_c$ better approximates a logarithmic function with increasing grid density. The importance of a grid in a landscape is conform the results of Alberti et al. [14], who differentiates roads as key stressors in a landscape.

In accordance to the findings of Shields and Tague [19], the metric EIA was found to have a large influence on the catchment's hydrology. EIA was modeled for each grid scenario and each degree

of perviousness separately (Figure 8). The steeper slope of the linear models at low pervious surface percentages means that in a strongly urbanized landscape with a lot of impervious surface, even a small change in EIA can lead to a dramatic increase in runoff. Landscapes with higher percentages of pervious area, on the other hand, are better buffered against an increase in $F_c$ when EIA rises.

The lesser performance of the EIA models at low degrees of pervious surface (Table 3) are due to the non-linear behavior of EIA. This non-linearity is corrected with the addition of the variable representing the number of pervious clusters ($N_{PC}$), that exhibits a strongly negative correlation with $F_c$ at 20% pervious surface (Table 2). The implication is that at high degrees of imperviousness, when few pervious spots are available for runoff to infiltrate, the spatial mixture of urban green becomes increasingly important and a stronger mixture of urban green is preferred. At higher degrees perviousness, there is sufficient area for runoff to infiltrate and disrupt the connectivity of impervious surfaces that enhances $F_c$. The added effect of spatial arrangement of urban green is negligible in these cases. The spatial arrangement of different urban green types thus only becomes important in highly urbanized landscapes with a lot of impervious surfaces.

*4.2. Optimization*

The sharp increase in $F_c$ from the Reference to the Point Breeze scenario (Figure 10) shows that a drainage system is a very efficient means to transport runoff water to the catchments' outlet. At higher grid densities, such as in the Point Breeze scenario, almost all runoff reaches the catchments' outlet, regardless of the percentage of pervious surface present. At lower grid densities, lower values of $F_c$ are observed because a smaller percentage of impervious surface is connected to the grid. More runoff flows into pervious areas where it can infiltrate. Depending on where the impervious surface is located relative to the outlet, produced runoff has to flow past patches of pervious surface. In the Irvine scenario and to a lesser extent in the Concord- and Point Breeze scenario, $F_c$ decreases until the amount of pervious area is 40%. More pervious area and a less dense grid thus means the possibility is higher that patches are isolated and not connected to the grid. After 40% pervious surface we see an increase in $F_c$ for the Irvine-, Concord-, and Point Breeze scenario. The reason for this is that in the higher percentage pervious scenarios, a higher percentage of the impervious area is road grid, which has a very high runoff potential. Most water that directly falls on road surface reaches the outlet, resulting in a higher $F_c$. When the road grid occupies a high percentage of the total impervious area, less impervious surface is available to disconnect from the drainage system. This will result in a high EIA and less potential disconnected impervious surface area. For example, in the 60% pervious scenario for the Concord gird, 18.9% of the remaining 40% impervious surface consists of road grid. All rain that falls on the road grid will be guided to the outlet. This means that only 21.1% of impervious surface remains to be reallocated and disconnected from the outlet. The potential reduction in $F_c$ will thus decrease with higher pervious surface percentages.

*4.3. Real World Implications*

The results of this virtual experiment reveal that the amount of runoff that reaches the catchments' outlet is highly dependent on the density of the street grid. The street grid in turn has a strong impact on the landscape metrics that were used in this study to measure the landscape pattern. The strong and consistent correlations between the metrics A, $N_{IC}$, and $STD_{IC}$ on the one hand and $F_c$ on the other showed that a fragmented, fine-grained landscape is able to absorb more runoff than a more clumped, course-grained landscape.

The metric EIA showed that the imperious area directly connected to the grid should be reduced if the goal is to let more water infiltrate in pervious spots in the city and the metrics $PX_{ICO}$ and $PX_{ICR}$ showed that large impervious clusters are best located far from the outlet or road to give runoff a better chance to infiltrate. The logarithmic models of the metric A (Figure 7) show that gaining a continuous reduction in $F_c$ comes with smaller efforts when $F_c$ is already low. The models of the metric EIA (Figure 8) show that, when pervious surface area is low, strong reductions in $F_c$ can be achieved

with only minor changes in EIA. The strongest reductions can be made when the reduction in EIA is accompanied with the creation of small impervious spots with a mixture of different urban green types (Models see Appendix B).

These findings advocate for the decentralization of the water system. By creating small pervious spots in the landscape that are disconnected from the drainage system runoff finds alternative spots to infiltrate, thereby relieving pressure on the traditional drainage system. These small pervious spots can have multiple uses: They can be traditional vegetation such as grass, shrubs, and trees that capture rainfall and let it infiltrate in the groundwater table or they could be water sensitive urban design (WSUD) systems such as rainwater tanks, bioretention systems, or swales that capture runoff for other purposes [42]. Recent remote sensing research found that cities around the world are made of a strongly fragmented mosaics of impervious- and pervious surface with often high to very high impervious area fractions [43]. Because of these high impervious surface area fractions, the construction of even a small amount of pervious spots can disconnect a substantial amount of impervious area, thereby strongly increasing natural infiltration and decreasing $F_c$. The positive correlations of the proximity indexes $PX_{ICO}$ and $PX_{ICR}$ with $F_c$ show that large impervious clusters are best located far from the outlet. In practice, large impervious spots such as parking areas are directly connected to the drainage system. Disconnecting these spots from the drainage system through the use of alternative management practices (e.g., the construction of a swale) provides the opportunity to greatly reduce $F_c$.

These potential reductions in $F_c$ increase with increasing grid density as well, which gives highly urbanized regions with a high grid density and low pervious surface percentage an extra favorable cost/benefit ratio to improve natural infiltration.

*4.4. Simplifications*

Due to the nature of virtual experiments, a lot of simplifications are made. These simplifications are acceptable because this study does not aim to make accurate predictions of runoff amounts but rather aims to unveil underlying patterns and relationships between variables. The assumption that road networks are a good substitute for the drainage system might not be the case everywhere. Moreover actual drainage systems can saturate, impeding outlet flow. The land use classes used in this study are a simplification of reality and the CN number assigned to them lies in the range of those of [35] but are a simplification nonetheless because they were not designed to differentiate between different urban green classes. By changing the CN numbers, the values of $F_c$ will undoubtedly change but the general patterns and relationships between the landscape metrics and $F_c$ found in this study will remain. The CN hydrological model is event based, no time variable is used and the effect of rain intensity and time of concentration is not investigated. We hypothesize that, with increasing rain intensity, the effect of the pervious patches will diminish because rain will have less time to infiltrate [44]. We think that the general results found in this research however will remain valid. The question also remains to which extent the generated NLMs capture the spatial configuration of an actual urban environment. The strength of this virtual experiment lies in the fact that it scales down the complexity of an urban environment and only allows certain components to vary, thereby exposing underlying mechanisms that could otherwise be obscured by the complexity of the data.

Finally, this study attempts to expose the mechanisms underlying $F_c$ by creating NLMs and calculating landscape metrics to predict $F_c$. It is however situation dependent if a higher- or lower $F_c$ is preferred. Cities located in semi-arid environments such as Irvine and Concord likely would prefer a low $F_c$ to keep as much rainwater in the city as possible for uses such as purification for drinking water or to water urban vegetation [19]. Cities or neighborhoods located in humid- and wet environments such as Point Breeze that have no water shortage might prefer a high $F_c$, but even in these cases it could be interesting for these cities to capture runoff and utilize it for other purposes than to guide the water away from the city.

*4.5. Future Research*

Recommendations for follow up studies are to add layers of complexity to the model such as the use of more land use classes, a drainage network with a maximal transport capacity and adding a time variable to investigate variability of peak flows and volumes. Urban environments are characterized by linear- and polygonal structures that have specific shapes such as buildings and parking spaces. The effect of these structures on $F_c$ would be interesting to examine. A similar experimental design as the one used in this paper could be used to determine the effect of these structures on $F_c$. Instead of creating patterns through a pseudo-random noise function, a function that creates patterns with these specific shapes, altering their size and orientation, could be used. By adding complexity, the bridge with reductionist models can be closed through the development of 'middle path' models that capture the spatial characteristics of a landscape without the risk of over parameterization [45]. Remote sensing data has a high potential to serve as good quality data source to quantify runoff volumes in an urban catchment, thereby facilitating data input [46]. The insights gained in this study can contribute to more thoughtful city planning. Further research should also be directed towards existing urban landscapes where only minor changes can be made. The optimization technique used in this research can be adapted towards real landscapes where additional constraints towards land use allocation could be implemented. Urban planners often only have limited budget and space to allocate pervious spots such as trees and shrubs. The optimization technique presented this paper can help them determine the most optimal location to implement pervious spots to maximally reduce $F_c$. Moreover, the methods presented in this paper can provide valuable data for decision support methods such as Multi-Criteria Decision Analysis (MCDA) [47].

## 5. Conclusions

$F_c$ varies significantly with street/drainage system density and spatial land use configuration. A dense grid is very efficient in transporting runoff to the outlet, giving rainwater few opportunities to infiltrate and replenish the ground water table. Our results indicate that stronger reductions in $F_c$ are achievable in a landscape with a sparser street grid. A strongly fragmented, fine-grained landscape with equally sized impervious patches was found to be more efficient in absorbing runoff than a more coarse-grained landscape. Our results indicate that the mixture of urban green only becomes important in these highly urbanized landscapes. The landscape metrics A, $N_{IC}$, $STD_{IC}$, EIA, $PX_{ICO}$, and $PX_{ICR}$ showed strong correlations with $F_c$ in all scenarios across all levels of perviousness. Their correlation with $F_c$ increased with a denser drainage system. The relationship of the metrics A and EIA with $F_c$ was modelled. The models of the metric A showed that by reducing the average impervious cluster area, increasing reductions in $F_c$ are to be expected. The EIA models showed that decreasing the effective impervious area might be a more effective measure and that the strongest reductions in $F_c$ can be accomplished in strongly urbanized landscapes with small percentages of pervious area. This reduction in EIA can be achieved through alternative measures such as rain gardens or tree infiltration pits that decentralize water management and partly mitigate the negative consequences of the increasing degree of impervious area in urban environments. The optimization analysis showed that around 40% pervious surface area, the lowest values of Fc are found in the grid scenarios. The results in this study can be used to improve our understanding of landscape dynamics on runoff in urban environments. Recommendations for further research include adding layers of complexity to the model to gain more insight on the interaction between landscape configuration and runoff. The optimization technique used in this study could also be useful for decision making on where to best construct urban green areas in real urban landscapes.

**Author Contributions:** Conceptualization, V.S., B.V., M.H. and B.S.; Data curation, V.S.; Formal analysis, V.S.; Investigation, V.S.; Methodology, V.S.; Project administration, B.V., M.H. and B.S.; Software, V.S.; Supervision, B.V., M.H. and B.S.; Visualization, V.S.; Writing—original draft, V.S.; Writing—review and editing, V.S., B.V., M.H. and B.S.

**Funding:** This research was funded by Fonds Wetenschappelijk Onderzoek Vlaanderen (FWO), grant number FWO-B 12124.

**Acknowledgments:** V.S. would like to thank Roberto D'Ippolito for the provision and introduction to the software HEEDS|MDO 2019.

**Conflicts of Interest:** The authors declare no conflict of interest.

## Appendix A. Landscape Metrics Spearman Correlations with $p$ Values

*Appendix A.1. Reference Scenario (n = 400)*

**Table A1.** Correlation table Reference scenario.

| | **Pervious Fraction** | | | | | **$p$ Values** | | | |
|---|---|---|---|---|---|---|---|---|---|
| $F_c$ | 0.20 | 0.40 | 0.60 | 0.80 | $F_c$ | 0.20 | 0.40 | 0.60 | 0.80 |
| A | 0.40 | 0.76 | 0.78 | 0.35 | A | 0.0001 | <0.0001 | <0.0001 | 0.0004 |
| $N_{IC}$ | −0.41 | −0.76 | −0.78 | −0.35 | $N_{IC}$ | 0.0001 | <0.0001 | <0.0001 | 0.0004 |
| $N_{PC}$ | −0.74 | −0.63 | −0.45 | −0.12 | $N_{PC}$ | <0.0001 | <0.0001 | <0.0001 | 0.2535 |
| $STD_{IC}$ | 0.40 | 0.80 | 0.81 | 0.34 | $STD_{IC}$ | 0.0002 | <0.0001 | <0.0001 | 0.0005 |
| $M_{IC}$ | 0.66 | 0.50 | 0.36 | 0.30 | $M_{IC}$ | <0.0001 | <0.0001 | 0.0002 | 0.0025 |
| $L_{IC}$ | −0.09 | 0.69 | 0.77 | 0.30 | $L_{IC}$ | 0.4154 | <0.0001 | <0.0001 | 0.0027 |
| EIA | 0.66 | 0.82 | 0.88 | 0.80 | EIA | <0.0001 | <0.0001 | <0.0001 | <0.0001 |
| $FP_{OIP}$ | −0.43 | −0.36 | −0.39 | −0.64 | $FP_{OIP}$ | <0.0001 | 0.0002 | <0.0001 | <0.0001 |
| $FP_{MICO}$ | 0.39 | 0.24 | 0.39 | −0.18 | $FP_{MICO}$ | 0.0002 | 0.0149 | <0.0001 | 0.0677 |
| $FP_{LICO}$ | −0.18 | −0.19 | −0.24 | −0.38 | $FP_{LICO}$ | 0.1058 | 0.0579 | 0.0153 | <0.0001 |
| $PX_{ICO}$ | 0.38 | 0.73 | 0.78 | 0.35 | $PX_{ICO}$ | 0.0003 | <0.0001 | <0.0001 | 0.0003 |
| $PX_{LICO}$ | 0.59 | 0.72 | 0.88 | 0.53 | $PX_{LICO}$ | <0.0001 | <0.0001 | <0.0001 | <0.0001 |

*Appendix A.2. Irvine scenario (n = 400)*

**Table A2.** Correlation table Irvine scenario.

| | **Pervious Fraction** | | | | | **$p$ Values** | | | |
|---|---|---|---|---|---|---|---|---|---|
| $F_c$ | 0.20 | 0.40 | 0.60 | 0.80 | $F_c$ | 0.20 | 0.40 | 0.60 | 0.80 |
| A | 0.69 | 0.90 | 0.93 | 0.93 | A | <0.0001 | <0.0001 | <0.0001 | <0.0001 |
| $N_{IC}$ | −0.69 | −0.90 | −0.93 | −0.93 | $N_{IC}$ | <0.0001 | <0.0001 | <0.0001 | <0.0001 |
| $N_{PC}$ | −0.78 | −0.60 | −0.38 | −0.15 | $N_{PC}$ | <0.0001 | <0.0001 | <0.0001 | 0.1244 |
| $STD_{IC}$ | 0.68 | 0.93 | 0.96 | 0.96 | $STD_{IC}$ | <0.0001 | <0.0001 | <0.0001 | <0.0001 |
| $M_{IC}$ | −0.01 | −0.08 | 0.05 | 0.21 | $M_{IC}$ | 0.9028 | 0.4062 | 0.6477 | 0.0338 |
| $L_{IC}$ | −0.03 | 0.88 | 0.92 | 0.94 | $L_{IC}$ | 0.7872 | <0.0001 | <0.0001 | <0.0001 |
| EIA | 0.79 | 0.97 | 0.97 | 0.97 | EIA | <0.0001 | <0.0001 | <0.0001 | <0.0001 |
| $FP_{OIP}$ | −0.05 | −0.08 | 0.01 | 0.15 | $FP_{OIP}$ | 0.6187 | 0.4364 | 0.9564 | 0.1467 |
| $FP_{RIP}$ | −0.08 | 0.88 | −0.24 | −0.33 | $FP_{RIP}$ | 0.4306 | <0.0001 | 0.0161 | 0.0009 |
| $FP_{MICO}$ | 0.10 | −0.08 | −0.06 | 0.06 | $FP_{MICO}$ | 0.3194 | 0.4036 | 0.5678 | 0.5223 |
| $FP_{MICR}$ | −0.45 | −0.29 | −0.83 | −0.75 | $FP_{MICR}$ | <0.0001 | 0.0038 | <0.0001 | <0.0001 |
| $FP_{LICO}$ | 0.01 | −0.20 | 0.02 | 0.17 | $FP_{LICO}$ | 0.9468 | 0.0442 | 0.8733 | 0.0841 |
| $FP_{LICR}$ | 0.22 | 0.04 | −0.19 | 0.14 | $FP_{LICR}$ | 0.0288 | 0.705 | 0.062 | 0.1788 |
| $PX_{ICO}$ | 0.64 | 0.88 | 0.93 | 0.94 | $PX_{ICO}$ | <0.0001 | <0.0001 | <0.0001 | <0.0001 |
| $PX_{ICR}$ | 0.70 | 0.89 | 0.93 | 0.93 | $PX_{ICR}$ | <0.0001 | <0.0001 | <0.0001 | <0.0001 |
| $PX_{LICO}$ | 0.21 | 0.67 | 0.89 | 0.89 | $PX_{LICO}$ | 0.0385 | <0.0001 | <0.0001 | <0.0001 |
| $PX_{LICR}$ | 0.07 | 0.71 | 0.84 | 0.85 | $PX_{LICR}$ | 0.478 | <0.0001 | <0.0001 | <0.0001 |

*Appendix A.3. Concord scenario (n = 300)*

**Table A3.** Correlation table Concord scenario.

| | **Pervious Fraction** | | | | | ***p* Values** | | | |
|---|---|---|---|---|---|---|---|---|---|
| $F_c$ | 0.20 | 0.40 | 0.60 | 0.80 | $F_c$ | 0.20 | 0.40 | 0.60 | 0.80 |
| A | 0.85 | 0.95 | 0.94 | | A | <0.0001 | <0.0001 | <0.0001 | |
| $N_{IC}$ | −0.85 | −0.95 | −0.94 | | $N_{IC}$ | <0.0001 | <0.0001 | <0.0001 | |
| $N_{PC}$ | −0.77 | −0.56 | −0.27 | | $N_{PC}$ | <0.0001 | <0.0001 | 0.0064 | |
| $STD_{IC}$ | 0.85 | 0.96 | 0.93 | | $STD_{IC}$ | <0.0001 | <0.0001 | <0.0001 | |
| $M_{IC}$ | 0.83 | 0.80 | 0.83 | | $M_{IC}$ | <0.0001 | <0.0001 | <0.0001 | |
| $L_{IC}$ | 0.39 | 0.88 | 0.86 | | $L_{IC}$ | <0.0001 | <0.0001 | <0.0001 | |
| EIA | 0.80 | 0.98 | 0.98 | | EIA | <0.0001 | <0.0001 | <0.0001 | |
| $FP_{OIP}$ | −0.11 | −0.07 | 0.06 | | $FP_{OIP}$ | 0.2782 | 0.5199 | 0.5329 | |
| $FP_{RIP}$ | 0.04 | −0.09 | −0.14 | | $FP_{RIP}$ | 0.7095 | 0.3545 | 0.1546 | |
| $FP_{MICO}$ | −0.03 | 0.16 | 0.30 | | $FP_{MICO}$ | 0.7619 | 0.1212 | 0.0022 | |
| $FP_{MICR}$ | 0.40 | −0.69 | −0.88 | | $FP_{MICR}$ | <0.0001 | <0.0001 | <0.0001 | |
| $FP_{LICO}$ | −0.18 | 0.07 | 0.08 | | $FP_{LICO}$ | 0.081 | 0.5001 | 0.4386 | |
| $FP_{LICR}$ | 0.05 | 0.19 | 0.29 | | $FP_{LICR}$ | 0.6363 | 0.0599 | 0.0034 | |
| $PX_{ICO}$ | 0.75 | 0.94 | 0.94 | | $PX_{ICO}$ | <0.0001 | <0.0001 | <0.0001 | |
| $PX_{ICR}$ | 0.85 | 0.94 | 0.93 | | $PX_{ICR}$ | <0.0001 | <0.0001 | <0.0001 | |
| $PX_{LICO}$ | 0.27 | 0.67 | 0.79 | | $PX_{LICO}$ | 0.0073 | <0.0001 | <0.0001 | |
| $PX_{LICR}$ | 0.38 | 0.48 | 0.63 | | $PX_{LICR}$ | <0.0001 | <0.0001 | <0.0001 | |

*Appendix A.4. Point Breeze scenario (n = 300)*

**Table A4.** Correlation table Point Breeze scenario.

| | **Pervious Fraction** | | | | | ***p* Values** | | | |
|---|---|---|---|---|---|---|---|---|---|
| **$F_c$** | **0.20** | **0.40** | 0.60 | 0.80 | $F_c$ | 0.20 | 0.40 | 0.60 | 0.80 |
| A | 0.93 | 0.97 | 0.97 | | A | <0.0001 | <0.0001 | <0.0001 | |
| $N_{IC}$ | −0.93 | −0.97 | −0.97 | | $N_{IC}$ | <0.0001 | <0.0001 | <0.0001 | |
| $N_{PC}$ | −0.77 | −0.50 | −0.13 | | $N_{PC}$ | <0.0001 | <0.0001 | 0.1906 | |
| $STD_{IC}$ | 0.88 | 0.93 | 0.92 | | $STD_{IC}$ | <0.0001 | <0.0001 | <0.0001 | |
| $M_{IC}$ | 0.83 | 0.91 | 0.88 | | $M_{IC}$ | <0.0001 | <0.0001 | <0.0001 | |
| $L_{IC}$ | 0.66 | 0.75 | −0.03 | | $L_{IC}$ | <0.0001 | <0.0001 | 0.777 | |
| EIA | 0.84 | 0.99 | 0.99 | | EIA | <0.0001 | <0.0001 | <0.0001 | |
| $FP_{OIP}$ | 0.04 | 0.01 | 0.07 | | $FP_{OIP}$ | 0.6899 | 0.9513 | 0.4646 | |
| $FP_{RIP}$ | 0.10 | −0.16 | −0.48 | | $FP_{RIP}$ | 0.2996 | 0.1203 | <0.0001 | |
| $FP_{MICO}$ | −0.09 | −0.10 | 0.04 | | $FP_{MICO}$ | 0.3504 | 0.3365 | 0.686 | |
| $FP_{MICR}$ | 0.79 | 0.54 | −0.21 | | $FP_{MICR}$ | <0.0001 | <0.0001 | 0.0356 | |
| $FP_{LICO}$ | −0.15 | 0.08 | 0.06 | | $FP_{LICO}$ | 0.1359 | 0.4532 | 0.5624 | |
| $FP_{LICR}$ | −0.25 | −0.54 | 0.01 | | $FP_{LICR}$ | 0.0122 | <0.0001 | 0.932 | |
| $PX_{ICO}$ | 0.89 | 0.96 | 0.97 | | $PX_{ICO}$ | <0.0001 | <0.0001 | <0.0001 | |
| $PX_{ICR}$ | 0.90 | 0.97 | 0.96 | | $PX_{ICR}$ | <0.0001 | <0.0001 | <0.0001 | |
| $PX_{LICO}$ | 0.17 | 0.22 | 0.71 | | $PX_{LICO}$ | 0.0958 | 0.0254 | <0.0001 | |
| $PX_{LICR}$ | 0.67 | 0.59 | 0.72 | | $PX_{LICR}$ | <0.0001 | <0.0001 | <0.0001 | |

# Appendix B

**Table A5.** Parameter estimates of variable A with Std. Error and 95% confidence intervals (Table 3 and Figure 7).

| Parameter | Group | Estimate | Std Error | Prob > \|t\| | Lower 95% | Upper 95% |
|-----------|-------|----------|-----------|-----------|-----------|-----------|
| a1 | Reference | 0.0399599 | 0.017786 | 0.0252 | 0.0049899 | 0.0749298 |
| b1 | Reference | 0.0620165 | 0.003245 | <0.0001 | 0.055637 | 0.068396 |
| a2 | Irvine | 0.4321842 | 0.012495 | <0.0001 | 0.4076191 | 0.4567494 |
| b2 | Irvine | 0.0686067 | 0.002824 | <0.0001 | 0.0630545 | 0.074159 |
| a3 | Concord | 0.7302278 | 0.00613 | <0.0001 | 0.7181638 | 0.7422918 |
| b3 | Concord | 0.0471453 | 0.001641 | <0.0001 | 0.0439159 | 0.0503748 |
| a4 | Point Breeze | 0.9120371 | 0.002328 | <0.0001 | 0.9074561 | 0.916618 |
| b4 | Point Breeze | 0.0178047 | 0.000689 | <0.0001 | 0.0164488 | 0.0191605 |

**Table A6.** Parameter estimates of variable EIA with Std. Error and 95% confidence intervals (Table 3 and Figures 8 and 9).

| Parameter | Group | Estimate | Std Error | Prob > \|t\| | Lower 95% | Upper 95% |
|-----------|-------|----------|-----------|-----------|-----------|-----------|
| a11 | Reference | 0.1452904 | 0.006163 | <0.0001 | 0.1331733 | 0.1574075 |
| b11 | Reference | 0.853726 | 0.018431 | <0.0001 | 0.8174878 | 0.8899642 |
| a21 | Irvine | 0.1549416 | 0.04437 | 0.0007 | 0.0668905 | 0.2429927 |
| b21 | Irvine | 0.8133143 | 0.056582 | <0.0001 | 0.7010291 | 0.9255995 |
| a22 | Irvine | 0.2409955 | 0.009794 | <0.0001 | 0.2215605 | 0.2604306 |
| b22 | Irvine | 0.7599255 | 0.016019 | <0.0001 | 0.7281361 | 0.7917148 |
| a23 | Irvine | 0.3676387 | 0.006727 | <0.0001 | 0.354289 | 0.3809884 |
| b23 | Irvine | 0.6365934 | 0.013302 | <0.0001 | 0.610197 | 0.6629898 |
| a24 | Irvine | 0.6111595 | 0.005233 | <0.0001 | 0.6007753 | 0.6215436 |
| b24 | Irvine | 0.3750707 | 0.012455 | <0.0001 | 0.3503533 | 0.3997881 |
| a31 | Concord | 0.2178195 | 0.054153 | 0.0001 | 0.1103556 | 0.3252834 |
| b31 | Concord | 0.7662501 | 0.059585 | <0.0001 | 0.6480046 | 0.8844955 |
| a32 | Concord | 0.4683952 | 0.006963 | <0.0001 | 0.4545781 | 0.4822122 |
| b32 | Concord | 0.5280021 | 0.008753 | <0.0001 | 0.5106317 | 0.5453725 |
| a33 | Concord | 0.7032752 | 0.003769 | <0.0001 | 0.6957962 | 0.7107542 |
| b33 | Concord | 0.2892382 | 0.005362 | <0.0001 | 0.2785979 | 0.2998785 |
| a41 | Point Breeze | 0.172284 | 0.051962 | 0.0013 | 0.0691663 | 0.2754018 |
| b41 | Point Breeze | 0.8173187 | 0.05337 | <0.0001 | 0.7114079 | 0.9232295 |
| a42 | Point Breeze | 0.6223457 | 0.003494 | <0.0001 | 0.615411 | 0.6292803 |
| b42 | Point Breeze | 0.3702626 | 0.003788 | <0.0001 | 0.3627463 | 0.3777789 |
| a43 | Point Breeze | 0.8837642 | 0.001389 | <0.0001 | 0.8810072 | 0.8865211 |
| b43 | Point Breeze | 0.1116972 | 0.001614 | <0.0001 | 0.1084947 | 0.1148997 |
| a5 | All grids | 0.2456423 | 0.006088 | <0.0001 | 0.2336991 | 0.2575855 |
| b5 | All grids | 0.7823386 | 0.008998 | <0.0001 | 0.7646876 | 0.7999895 |

# Appendix C

EIA models of Table 3 and Figure 8 with 20% pervious surface area before and after addition of variable NPC.

**EIA models with 20% pervious surface area**

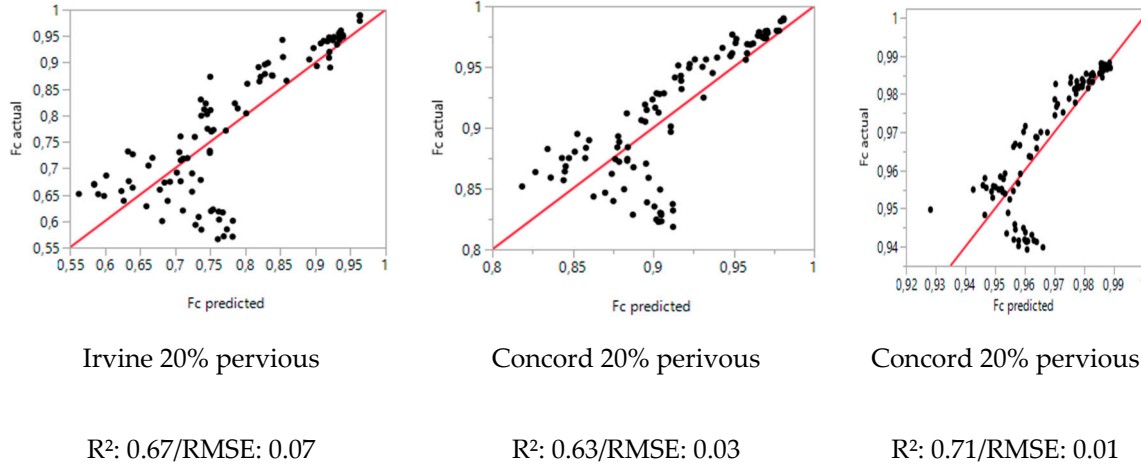

|  |  |  |
|---|---|---|
| Irvine 20% pervious | Concord 20% perivous | Concord 20% pervious |
| R²: 0.67/RMSE: 0.07 | R²: 0.63/RMSE: 0.03 | R²: 0.71/RMSE: 0.01 |

**Figure A1.** EIA models with 20% pervious surface.

**EIA models with 20% pervious area after addition of variable NPC**

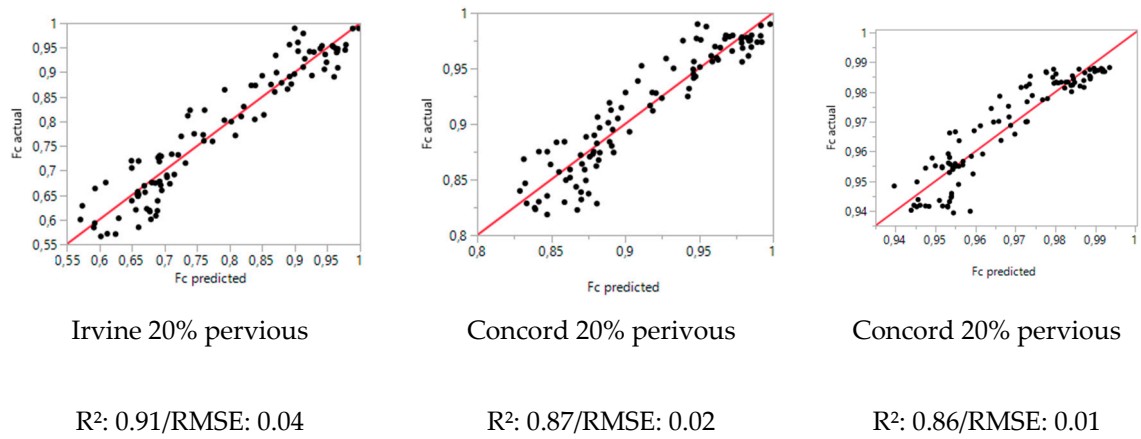

|  |  |  |
|---|---|---|
| Irvine 20% pervious | Concord 20% perivous | Concord 20% pervious |
| R²: 0.91/RMSE: 0.04 | R²: 0.87/RMSE: 0.02 | R²: 0.86/RMSE: 0.01 |

**Figure A2.** EIA models with 20% pervious area after addition of the variable NPC.

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
