# Peer review of "Urban Spatial Configuration and Functional Runoff Connectivity: Influence of Drainage Grid Density and Landscape Metrics"

_water, doi:10.3390/w11122661_

Round 1
Reviewer 1 Report
This is a very interesting paper. It is well written and properly organised.
I present the following comments that can help to improve the paper:
1. When starting a sentence with a reference, type the name of the authors first, e.g.: "[14] used flow path length..." to " Mayor et al. [14] used flow path length..."
2. Some references were not numbered and were not included in the list, e.g. Jacobson, 2011.
3. Check all abbreviations and symbols, e.g. sometimes the authors write Fc and others Fc.
4. Overall, Figures and Figure captions could be improved, e.g. Figure 2 should include a scale and the caption should include information on the area of each scenario; in Figure 6, axis labels should be outside the graphs.
5. Table 2 has two captions and information on the grid should be more prominent.
6. Formula of the EIA should be displayed.
7. L258-260: "The street grid and impervious surface produced 15.5 and 5.2 mm of runoff respectively, while the grass-, trees- and shrub patches could absorb 5, 13 and 28.7 mm of runoff before saturating, respectively." This sentence is confusing. The value of runoff production by impervious surfaces is related to the amount of rainfall. On the contrary, the absorption by pervious surfaces is not related to rainfall.
8. In Table 3, the Y in the functions should be replaced by Fc and the X in the logarithmic function should be replaced by A. Also, use a1 and b1 instead of a1 and b1, because of in the functions you have ai and bi.
9. Figures and Tables should appear after first mentioned in the text
10. What is the n in Figures 7-9.
Author Response
This is a very interesting paper. It is well written and properly organised.
I present the following comments that can help to improve the paper:
Authors: We would like to thank the reviewer for his positive feedback. We addressed his comments as good as possible.
When starting a sentence with a reference, type the name of the authors first, e.g.: "[14] used flow path length..." to " Mayor et al. [14] used flow path length..."
Authors: Done throughout the text as suggested by the reviewer.
Some references were not numbered and were not included in the list, e.g. Jacobson, 2011.
Authors: We rechecked all references and all of them are listed now in the references.
Check all abbreviations and symbols, e.g. sometimes the authors write Fcand others Fc.
Authors: Changed to Fc throughout the text.
Overall, Figures and Figure captions could be improved, e.g. Figure 2 should include a scale and the caption should include information on the area of each scenario; in Figure 6, axis labels should be outside the graphs.
Authors: We added a scale to the Figure 2 and included added this information in the caption. In Figure 6 the captions are moved outside the graphs.
Table 2 has two captions and information on the grid should be more prominent.
Authors: We removed the double caption. We are not sure what you mean with that the information on the grid should be more prominent.
Formula of the EIA should be displayed.
Authors: The EIA is calculated with the Flow Distance tool in arcpy. This tool calculates the flow distance to the nearest road for all impervious surface pixels. When the flow of an impervious pixel does not reach a road, this pixel is assigned a ‘No Data’ value. The ratio of impervious pixels where the flow reaches a road over all impervious pixels is the EIA. We added this explanation on L236-240.
L258-260: "The street grid and impervious surface produced 15.5 and 5.2 mm of runoff respectively, while the grass-, trees- and shrub patches could absorb 5, 13 and 28.7 mm of runoff before saturating, respectively." This sentence is confusing. The value of runoff production by impervious surfaces is related to the amount of rainfall. On the contrary, the absorption by pervious surfaces is not related to rainfall.
Authors: Sentence changed to:
‘During a rain event of 20 mm, the street grid and impervious surface produced 15.5 and 5.2 mm of runoff respectively, while no runoff was generated on the grass-, trees- and shrub patches. During and after the rain event, these patches could absorb an additional 5, 13 and 28.7 mm of runoff respectively from nearby impervious surfaces before saturating.’
In Table 3, the Y in the functions should be replaced by Fc and the X in the logarithmic function should be replaced by A. Also, use a1and b1 instead of a1 and b1, because of in the functions you have ai and bi.
Authors: Done.
Figures and Tables should appear after first mentioned in the text
Authors: We rechecked this and changed this where needed.
What is the n in Figures 7-9.
Authors: These are the number of NLM’s generated for each grid scenario. Because in the Concord and Point Breeze scenarios no NLM’s with 80% pervious surface are generated, these scenarios only have 300 NLM’s. We added this explanation in the figure captions.
Reviewer 2 Report
The study aims to present "Urban spatial configuration and functional runoff connectivity: Influence of drainage grid density and landscape metrics". This is an interesting topic for the Journal Water. However, I suggest some of the comments.
1. Introduction:
It will be better if the author could cite some of the most recent literature. i.e. Line 49
2. What are the limitations of this study?
3. Reference should be arranged according to the Water.
Author Response
The study aims to present "Urban spatial configuration and functional runoff connectivity: Influence of drainage grid density and landscape metrics". This is an interesting topic for the Journal Water. However, I suggest some of the comments.
Authors: We thank the reviewer for his feedback, see our responses to his comments below.
Introduction:
It will be better if the author could cite some of the most recent literature. i.e. Line 49
Authors: We assume the reviewer is referring to this line ‘ The conventional solution to these problems is to guide the excess water away from the city.’.
We now cite the meta-analysis of Zhang & Ariaratnam (2018), who explains the problem of stormwater in urbanized regions and proposes some mitigation strategies.
Zhang, P.; Ariaratnam, S.T. Meta-Analysis of Storm Water Impacts in Urbanized Cities Including Runoff Control and Mitigation Strategies. J. Sustain. Dev. 2018, 11, 27.
What are the limitations of this study?
Authors: Because of the simplifications made in creating the Neutral Landscape Models, the results of this paper should not be used to make precise predictions of runoff. The results expose underlying processes and landscape metrics that influence the amount of runoff that reaches the catchment’s outlet and help to improve our understanding of the urban hydrological balance. We elaborate on this in section ‘4.4. Simplifications’ and section ‘4.5. Future research’. We extended these sections with some additional thoughts.
Reference should be arranged according to the Water.
Authors: Done.
Reviewer 3 Report
It was difficult for me to understand the intention of the authors. Their manuscript contains results of in depth analysis of virtual (hypothetical) urban spatial configurations on functional runoff connectivity for three cases of street networks. The concept of the research seems to be appropriate and could be attractive for the city planners. However the assumptions underlying the methodology applied are questionable and thus obtained results unpractical.
The authors are aware of the limitations of their research therefore I do not understand why the authors decided not to consider in their analysis as constrains the urban landscape features which cannot be changed but influence the runoff to a large extent like rivers and buildings not to mention DEM. Instead only street network configurations was taken from real world and the location of impervious ‘pixels’ which could be a building, is subject to change in the neutral landscape model generation. In my opinion the number of scenarios should also be reduced by eliminating unrealistic scenarios e.g high ratio of pervious surface and coarse street network or low ratio of pervious surface and dense street network. If the authors decide to increase the number of constrains the reference scenario should be the present state of landscape (land cover) in selected cities.
In my opinion the same conclusions could be reached with less scenarios but more realistic.
Additional specific comments which could improve the manuscript are as follow:
Please remove titles of subsections from Introduction chapter. L65 The style of citation is incorrect and the reference is missing. L98-99 Please provide the answer to this question in Conclusions chapter. L114 Please do not use abbreviations (CN, Fc) before they are explained. Please modify the workflow presented in Fig. 1 to show the interdependence between the three parts. Remove ‘depicting the three main parts of the study’ from the title. L128 Please provide the dimensions (size) of the of the abstracted areas with grids. L137 Provide units for the axes and legend. L143 Please provide the real size of one pixel. L172 Again Fc is mentioned but was not explained before. L181 Please explain more clearly the term ‘cluster’. L193 The definition of ‘proximity’ is also unclear. I am wondering where in this research ‘Downward modelling approach’ was applied? L226 This is the most important section of the manuscript so the detailed description of runoff calculation should be provided. What is the formula for runoff in CN method? Why the chosen values of CN for different classes are provided with the precision of one decimal place e.g. 97.9 and not integral number. How the runoff is routed to the outlet? Also the only one equation provided for functional runoff connectivity (Fc) is unclear. Moreover please explain how the ‘Total outlet volume’ and ‘Total internal runoff production’ are calculated. Normally the first is measured data. L264 Delete (DOE). It was already extended in previous line. L296-298 This paragraph looks like second table caption. L335 Figure 6 repeats data from Table 2. Could it be removed? L352 Not clear what are Y and X in model functions. Please replace them with appropriate symbols. L644 Provide full names of all co-authors.
Comments for Editors
Taking into consideration my comments regarding totally unrealistic scenarios of the experiment and insufficient description of the runoff calculation my overall recommendation is:
Reconsider after major revision (control missing in some experiments)
Author Response
Suggestions for Authors
It was difficult for me to understand the intention of the authors. Their manuscript contains results of in depth analysis of virtual (hypothetical) urban spatial configurations on functional runoff connectivity for three cases of street networks. The concept of the research seems to be appropriate and could be attractive for the city planners. However the assumptions underlying the methodology applied are questionable and thus obtained results unpractical.
The authors are aware of the limitations of their research therefore I do not understand why the authors decided not to consider in their analysis as constrains the urban landscape features which cannot be changed but influence the runoff to a large extent like rivers and buildings not to mention DEM. Instead only street network configurations was taken from real world and the location of impervious ‘pixels’ which could be a building, is subject to change in the neutral landscape model generation. In my opinion the number of scenarios should also be reduced by eliminating unrealistic scenarios e.g high ratio of pervious surface and coarse street network or low ratio of pervious surface and dense street network. If the authors decide to increase the number of constrains the reference scenario should be the present state of landscape (land cover) in selected cities.
In my opinion the same conclusions could be reached with less scenarios but more realistic.
Authors: We appreciate the critical look of the reviewer on our paper. We have separated his general criticism in separate parts and formulated an answer to them below:
Reviewer: ‘The concept of the research seems to be appropriate and could be attractive for the city planners. However the assumptions underlying the methodology applied are questionable and thus obtained results unpractical.’
Authors: It was not the goal of this research to deliver readily applicable results for city planners. The goal was to determine landscape metrics that influence runoff and to examine their relation with runoff (more specifically the functional runoff connectivity). We clarify this in the last sentence of the abstract:
‘By using a downward hydrological modeling approach this research aims to bring more clarity to the underlying variables influencing Fc, rather than trying to generate realistic prediction values.’
And in section ‘4.4 Simplifications’:
‘These simplifications are acceptable because this study does not aim to make accurate predictions of runoff amounts but rather aims to unveil underlying patterns and relationships between variables’
‘The strength of this virtual experiment lies in the fact that it scales down the complexity of an urban environment and only allows certain components to vary, thereby exposing underlying mechanisms that could otherwise be obscured by the complexity of the data.’
We explain our reasoning to use this methodology in section ‘2.2.1 Downward modelling approach’.
We think the insights gained in this study are valuable to obtain better insight on how landscape pattern influences runoff. We do not think the results of this study can or should be readily applied to city planning. Nowhere did we state that the results of this experiment should be used for practical decision making concerning city planning but rather to increase our understanding of landscape-run off interactions. To do that, we recommend using a physically based whole water balance model such as WetSpa (https://www.vub.be/WetSpa/welcome.html).
Reviewer: The authors are aware of the limitations of their research therefore I do not understand why the authors decided not to consider in their analysis as constrains the urban landscape features which cannot be changed but influence the runoff to a large extent like rivers and buildings not to mention DEM. Instead only street network configurations was taken from real world and the location of impervious ‘pixels’ which could be a building, is subject to change in the neutral landscape model generation.
Authors: The goals of this study was to determine the influence of drainage grid density and landscape metrics on the functional runoff connectivity. We did not impose constraints on the landscape features besides the percentage pervious surface (20,40,60 or 80%) because this ensured that the pseudo-random Perlin Noise function could generate landscapes with a wide variety of landscape patterns. Not all of the generated landscape patterns will have realistic configurations nor do they need to. By creating this wide variety of landscape patterns, underlying relationships between landscape metrics and Fc are deduced that otherwise could be obscured by only using actual urban land use patterns. The reason we chose these three drainage grids was because of their marked differences in grid density, context depended landscape features like as the local topography weren’t our interest. By generating one identical DEM for all drainage grids we ensured that the relationships that we were interested in could be clearly observed. When the actual DEM of the grids was used for the study, these results could have been obscured by the local topography. So with our research design we tried to control for confounding factors. We do agree with the reviewer that it would be interesting to use the methodology developed in this paper to investigate the effect of certain urban forms such as buildings on the functional runoff connectivity. We added the following lines to ‘4.5 Future research’:
‘Urban environments are characterized by linear- and polygonal structures that have specific shapes such as buildings and parking spaces. The effect of these structures on Fc would be interesting to examine. A similar experimental design as the one used in this paper could be used to determine the effect of these structures on Fc. Instead of creating patterns through a pseudo-random noise function, a function that creates patterns with these specific shapes, altering their size and orientation, could be used.’
Reviewer: In my opinion the number of scenarios should also be reduced by eliminating unrealistic scenarios e.g high ratio of pervious surface and coarse street network or low ratio of pervious surface and dense street network. If the authors decide to increase the number of constrains the reference scenario should be the present state of landscape (land cover) in selected cities.
In my opinion the same conclusions could be reached with less scenarios but more realistic.
Authors: We applied the same reasoning here as we did with the neutral landscape model generation. We are of the opinion that in a conceptual study like this, that is focused on exposing underlying relationships between landscape metrics and Fc, not all scenarios need to be realistic. We think that clearer relationships can be obtained by generating a large number of neutral landscape models with a wide variety of landscape patterns, rather than only working with actual landscape patterns. A next step could be to do the same analysis with existing landscape patterns only and see to which extent the relationships found in this research change.
Additional specific comments which could improve the manuscript are as follow:
Please remove titles of subsections from Introduction chapter.
Authors: We think that by removing the subtitles ‘1.1. Context’, ‘1.2. Problem statement’ and ‘1.3. Research questions’ , this part of the text would become too long and difficult to read. Therefor we prefer to keep the subtitles in the introduction chapter.
L65 The style of citation is incorrect and the reference is missing.
Authors: We added the reference to the reference list and rechecked all references. We also checked that the correct style of citation is used throughout the manuscript.
L98-99 Please provide the answer to this question in Conclusions chapter.
Authors: we added the following text in the conclusion:
‘Fc varies significantly with street/drainage system density and spatial land use configuration. A dense grid is very efficient in transporting runoff to the outlet, giving rain water few opportunities to infiltrate and replenish the ground water table. Our results indicate that stronger reductions in Fc are achievable in a landscape with a sparser street grid’
And
‘The optimization analysis showed that around 40% pervious surface area, the lowest values of Fc are found in the grid scenarios.’
L114 Please do not use abbreviations (CN, Fc) before they are explained.
Authors: We added the meaning of the abbreviations in the caption of Figure 1 and mention that they will be explained in the remainder of the Material and Methods section.
Please modify the workflow presented in Fig. 1 to show the interdependence between the three parts. Remove ‘depicting the three main parts of the study’ from the title.
Authors: We added arrows between the three work modules to emphasize their interdependence and removed the requested text from the caption.
L128 Please provide the dimensions (size) of the of the abstracted areas with grids.
Authors: We added a scale to Figure 2 and mention the extent of the grid in the caption.
L137 Provide units for the axes and legend.
Authors: Done
L143 Please provide the real size of one pixel.
Authors: Done
L172 Again Fc is mentioned but was not explained before.
Authors: We changed Fc here to runoff and explain Fc later in section 2.2.2.
L181 Please explain more clearly the term ‘cluster’.
Authors: We added ‘A cluster or patch is a basic element or unit that makes up a landscape [26]. In this study a cluster is defined as a unit of uniform land cover.’
L193 The definition of ‘proximity’ is also unclear.
Authors: We explained this landscape metric in more detail in the text:
‘The proximity index (PX) gives an indication of the relative importance of a cluster in relation to other clusters. It is calculated as the ratio of the impervious clusters’ sizes to the clusters’ flow path to the outlet or road respectively. In this research the PX is used to describe the contribution of an impervious cluster to the amount of runoff that reaches the outlet. For example, when a small impervious cluster close to the outlet has the same contribution to the amount of runoff that reaches the outlet as larger impervious cluster that is further away from the outlet, their PX is the same.’
This variable is also used in other studies such as:
Van Nieuwenhuyse, B.H.J.; Antoine, M.; Wyseure, G.; Govers, G. Pattern-process relationships in surface hydrology: Hydrological connectivity expressed in landscape metrics. Hydrol. Process. 2011, 25, 3760–3773.
I am wondering where in this research ‘Downward modelling approach’ was applied?
Authors: The downward modelling approach starts from a simple framework and is better suited to provide insights in relationships between variables. In this research we use a simple landscape generating framework with a select number of land covers. We simplify or eliminate possible confounding factors such as for example local topography (one DEM for all drainage grids) and spatial- and temporal variability of rainfall (we assume no time or spatial rainfall variability). This allows us to find clearer relationships between the variables that we are interested in.
L226 This is the most important section of the manuscript so the detailed description of runoff calculation should be provided. What is the formula for runoff in CN method? Why the chosen values of CN for different classes are provided with the precision of one decimal place e.g. 97.9 and not integral number. How the runoff is routed to the outlet? Also the only one equation provided for functional runoff connectivity (Fc) is unclear. Moreover please explain how the ‘Total outlet volume’ and ‘Total internal runoff production’ are calculated. Normally the first is measured data.
Authors: The CN runoff model is a widely used and accepted model for overland runoff calculations and the model basis and equations are well documented. To limit the length of our paper, we decided not to explicitly describe the formulas for runoff calculation but chose to refer to the original papers that describe the model formulas in detail (see section 2.2.2.). The original CN model was developed for agricultural purposes and therefor might not be fully applicable to a more heterogeneous urban environment. We used the methodology developed by Lim et al. (2006) to convert the original CN numbers to a value with one decimal place that is more adapted to an urban environment. We also refer to his paper for the more recent equation for the intial abstraction calculation of Ia= 0.05*S instead of Ia = 0.20*S.
Lim, K.J.; Engel, B.A.; Muthukrishnan, S.; Harbor, J. Effects of initial abstraction and urbanization on estimated runoff using CN technology. J. Am. Water Resour. Assoc. 2006, 42, 629–643.
Concerning the formula of Fc:
The ‘total outlet volume’ is the runoff volume that the CN model predicts that reaches the outlet, while the ‘total internal runoff production’ is the total volume of runoff that is produced on the impervious surface. Because a portion of the produced runoff will infiltrate in pervious areas, not all runoff that is produced will reach the outlet. The Fc hence will always have a dimensionless value between 0 and 1.
L264 Delete (DOE). It was already extended in previous line.
Authors: Done
L296-298 This paragraph looks like second table caption.
Authors: This caption of this table was indeed double, we changed this.
L335 Figure 6 repeats data from Table 2. Could it be removed?
Authors: We prefer to keep Figure 6 because it visualizes the relevant data of Table 2 that will be used for further modelling.
L352 Not clear what are Y and X in model functions. Please replace them with appropriate symbols.
Authors: Y is the dependent variable (Fc) and X are the independent variables (A and EIA). We changed this in the table.
L644 Provide full names of all co-authors.
Authors: Done.
Comments for Editors
Taking into consideration my comments regarding totally unrealistic scenarios of the experiment and insufficient description of the runoff calculation my overall recommendation is:
Reconsider after minor revision (control missing in some experiments)
Authors: We hope we clarified the questions the reviewer had regarding the goal and setup of the experiment and hope that he reconsiders his opinion of the manuscript. If the reviewer has further questions, we remain available for further clarification.
Reviewer 4 Report
First of all, I appreciate the overall logic and clarity of the reasoning in this paper. I like the experimental design of your study as it improves the understanding of how the urban water regime depends on land use.
However, this article will need some work to be done.
The landscape metrics you use are not a common meaning metrics in the sense of Mc Garigal and Marks 1994 (https://www.umass.edu/landeco/pubs/mcgarigal.marks.1995.pdf). Your study (and others on the subject) uses indicators adopted for hydrological applications. In this context, the use of the name landscape indicators may be misleading. Comment on this is needed.
Detailed comments:
Line 79: I don't understand what exactly you mean by "...the landscapes where the land use classes were more spatially distributed instead of uniform."
The terms 'uniform' and 'more spatially distributed' do not define the spatial configuration of the land use classes.
Line 114: The abbreviations used in the diagram should be explained in the drawing caption. Throughout this article, they are introduced here for the first time, and the text describing the methodology is further away.
Line 128: The information about the area of the examples should be provided before the figure 3, together with the other text describing the patterns.
Line 172: The abbreviation "Fc" is introduced here for the first time and should be explained.
Lines 175-177: Picture 4 is poorly readable, map enlargement and better green colour differentiation should improve communication.
Lines 183-199: Consider creating a table combining the contents of Table 1 and the text below. This will help you better explain the methodology for calculating landscape metrics.
Line 283: I do not understand the term "landscape design" used in this context. It is usually used to relate to spatial planning and landscape architecture. Did you mean patterns?
Line 296: First place the text and then the tables. The chapter must not start with table captions.
Line 350-362: Parameters used in the table should be described in the table caption. The text describing the results shown in the table should be located before it.
Line 432: A sentence should not start with a citation.
Lines 554-557: I do not agree that the presented method can solely determine the optimal location of previous spots. It can provide important data for decision support methods such as Multi-Criteria Decision Analysis, MCDA (e.g. Malczewski 2006).
Lines 554-557: Consider merging the 'Future Research' and 'Conclusion' chapters, as some content repeats.
Although I do not undertake a comprehensive evaluation of the language side of the text, I recommend checking it for editing errors and prepositions, commas, hyphens, uniform way of citation, etc.
Author Response
First of all, I appreciate the overall logic and clarity of the reasoning in this paper. I like the experimental design of your study as it improves the understanding of how the urban water regime depends on land use.
However, this article will need some work to be done.
Authors: We would like to thank the reviewer for his constructive feedback and addressed his comments as good as we could.
The landscape metrics you use are not a common meaning metrics in the sense of Mc Garigal and Marks 1994 (https://www.umass.edu/landeco/pubs/mcgarigal.marks.1995.pdf). Your study (and others on the subject) uses indicators adopted for hydrological applications. In this context, the use of the name landscape indicators may be misleading. Comment on this is needed.
Authors: Landscape metrics or indicators were originally developed from an ecological perspective, as explained by the FRAGSTAT paper the reviewer refers to. The paper however states there is room for different applications besides ecological studies. We cite from the paper:
P4: ‘The important point is that a landscape is not necessarily defined by its size; rather, it is defined by an interacting mosaic of patches relevant to the phenomenon under consideration (at any scale)’
In our paper, we use the concept of landscape metrics from an hydrological perspective and the phenomenon under consideration is runoff (specifically Fc). We define the landscape metrics from this point of view in Table 1. For clarity, we added the text: L216-219 ‘.Landscape metrics were originally developed to measure the ecological impacts of landscape changes [26] but have since been applied to several other domains such as water quality assessment, urban sprawl detection and even to visual landscape appreciation [27].’
We cite the paper of Evelin et al. (2009) who discusses the different uses of landscape metrics in different disciplines.
Evelin, U.; Marc, A.; Juri, R.; Riho, M.; M Landscape Metrics and Indices: An Overview of Their Use in Landscape Research. Living Rev. Landsc. Res. 2009, 3, 1–28.
Detailed comments:
Line 79: I don't understand what exactly you mean by "...the landscapes where the land use classes were more spatially distributed instead of uniform."
The terms 'uniform' and 'more spatially distributed' do not define the spatial configuration of the land use classes.
Authors: We agree that the terminology used in this sentence is confusing. We changed the sentence to:
‘By creating landscapes with different spatial land use configurations and comparing water- and erosion fluxes, they found that discharge was greater for the landscapes where the land use classes were coarsely clustered. ‘
Line 114: The abbreviations used in the diagram should be explained in the drawing caption. Throughout this article, they are introduced here for the first time, and the text describing the methodology is further away.
Authors: We added the meaning of the abbreviations to the caption and mention that they will be explained in the remainder of the Material and Methods section.
Line 128: The information about the area of the examples should be provided before the figure 3, together with the other text describing the patterns.
Authors: We added a scale to Figure 2 for area reference. The sentence ‘ Four km² of these urban grids were downloaded from www.openstreetmap.org with the Osmnx python plugin [21]’ is moved before Figure 2. The caption of Figure 3 is extended to explain the color pattern.
Line 172: The abbreviation "Fc" is introduced here for the first time and should be explained.
Authors: We changed ‘Fc’ here to ‘runoff’ because the explanation of Fc is done later and there is no need to introduce this term here yet. The first mention of Fc is now in section 2.2.2, where this variable is explained.
Lines 175-177: Picture 4 is poorly readable, map enlargement and better green colour differentiation should improve communication.
Authors: We put the last 2 panels below the first 2 to obtain larger images. We hope this improves the readability. We prefer to keep the green colors used in this image to illustrate that we are generating different types of green cover.
Lines 183-199: Consider creating a table combining the contents of Table 1 and the text below. This will help you better explain the methodology for calculating landscape metrics.
Authors: In the first draft of this paper, we created the table like the reviewer suggests. The table however contained a lot of text, which decreased the readability. We prefer the more simple table with some of the landscape metrics explained in the text below.
Line 283: I do not understand the term "landscape design" used in this context. It is usually used to relate to spatial planning and landscape architecture. Did you mean patterns?
Authors: Yes, we mean pattern and changed the term ‘landscape designs’ to ‘landscape patterns’ in this section.
Line 296: First place the text and then the tables. The chapter must not start with table captions.
Authors: We added explanatory text before the table.
Line 350-362: Parameters used in the table should be described in the table caption. The text describing the results shown in the table should be located before it.
Authors: Parameter descriptions added in the caption. The text describing the results is moved above the table.
Line 432: A sentence should not start with a citation.
Authors: Following the suggestions of reviewer 1, we started the sentence with ‘Mayor et al. [16] used …’ instead of ‘[16] used…’.
Lines 554-557: I do not agree that the presented method can solely determine the optimal location of previous spots. It can provide important data for decision support methods such as Multi-Criteria Decision Analysis, MCDA (e.g. Malczewski 2006).
Authors: We agree and added this to the ‘Future Research’ section.
Lines 554-557: Consider merging the 'Future Research' and 'Conclusion' chapters, as some content repeats.
Authors: To limit the size of the conclusion and because we extended the ‘Future Research’ section with some additional thoughts, we decided to keep both sections.
Although I do not undertake a comprehensive evaluation of the language side of the text, I recommend checking it for editing errors and prepositions, commas, hyphens, uniform way of citation, etc.